# LEARNING LARGE SKILLSETS IN STOCHASTIC SETTINGS WITH EMPOWERMENT

## ABSTRACT

General purpose agents need to be able to execute large skillsets in stochastic settings. Given that the mutual information between skills and states measures the number of distinct skills in a skillset, a compelling objective for learning a diverse skillset is to find the skillset with the largest mutual information between skills and states. The problem is that the two main unsupervised approaches for maximizing this mutual information objective, Empowerment-based skill learning and Unsupervised Goal-Conditioned Reinforcement Learning, only maximize loose lower bounds on the mutual information, which can impede diverse skillset learning. We propose a new empowerment objective, Skillset Empowerment, that maximizes a tighter bound on the mutual information between skills and states. For any proposed skillset, the tighter bound on mutual information is formed by replacing the posterior distribution of the proposed skillset with a variational distribution that is conditioned on the proposed skillset and trained to match the posterior of the proposed skillset. Maximizing our mutual information lower bound objective is a bandit problem in which actions are skillsets and the rewards are our mutual information objective, and we optimize this bandit problem with a new actor-critic architecture. We show empirically that our approach is able to learn large abstract skillsets in stochastic domains, including ones with high-dimensional observations, in contrast to existing approaches.

## 1 INTRODUCTION

General purpose agents that operate in the real world will need to be able to execute a large set of skills in highly stochastic settings. A futuristic household robot, for instance, will need to execute the vast number of skills involved in household chores like cooking and cleaning while the human members of the household may be moving as well as conversing with each other and the robot in seemingly random ways. Even the simple act of the robot moving its head to look in different directions will produce unpredictable outcomes as relevant objects to the robot may appear in unexpected places. An appealing approach for learning diverse skillsets, regardless of the level of randomness in the domain, is to find the skillset with the largest mutual information between skills and skill-terminating states because this mutual information measures the number of distinct skills in a skillset.

The problem is that the two most popular approaches for optimizing the mutual information between skills and states, Empowerment-based skill learning (Gregor et al., 2016; Eysenbach et al., 2018; Achiam et al., 2018; Choi et al., 2021; Sharma et al., 2019) and Unsupervised Goal-Conditioned Reinforcement Learning (GCRL) (Ecoffet et al., 2019; Mendonca et al., 2021; Nair et al., 2018; Pong et al., 2019; Campos et al., 2020; Pitis et al., 2020; Held et al., 2017; McClinton et al., 2021; Held et al., 2017; Kim et al., 2023), only maximize a loose lower bound of the mutual information between skills and states. The loose bound means agents are not able to accurately measure the diversity of candidate skillets, which in turn makes it difficult to find a diverse skillset. In both existing empowerment and unsupervised GCRL approaches, this loose lower bound on mutual information for a candidate skillset is formed by replacing the true posterior probability that computes the probability of a skill given the skill-terminating state and the candidate skillset with a potentially very different distribution. Existing empowerment approaches replace the true posterior of the candidate skillset with another distribution trained to match the true posterior of the *current* skillset, which may have significant differences with the candidate skillset. This can create a loose lower bound on

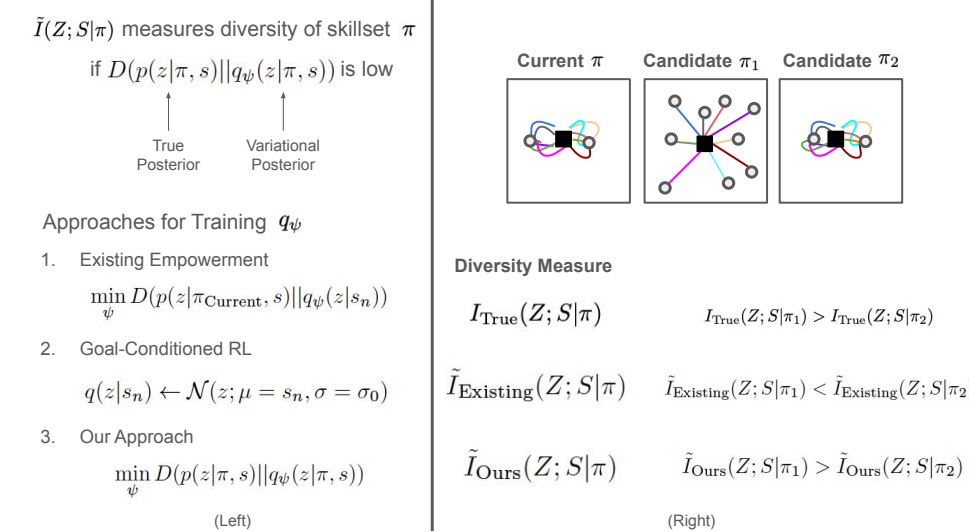

Figure 1: (Left) Overview of the problematic ways existing empowerment and GCRL train the variational posterior $q_\psi$. The variational mutual information $\tilde{I}(Z; S|\pi)$ between skills $z$ and states $s$ measures the diversity of skillset $\pi$ accurately if $q_\psi$ is similar to the true posterior $p(z|\pi, s)$ (i.e., the "distance" between the distributions $D(p||q_\phi)$ is small). But existing empowerment methods train the variational posterior, $q_\psi$, for a candidate skillset $\pi$ to match the posterior of the current skillset $p(z|\pi_{\text{Current}}, s_n)$, even when $\pi_{\text{Current}}$ is different than the candidate skillset $\pi$. GCRL sets the variational posterior to a fixed Gaussian centered at the skill-terminating state, which also does not depend on the skillset candidate $\pi$. In contrast, our approach trains $q_\psi$ to minimize its difference with the posterior of the candidate skillset $\pi$, improving the accuracy of $\tilde{I}(Z; S|\pi)$. (Right) Two candidate skillsets $\pi_1$ and $\pi_2$ are compared using different measures of diversity. $\pi_1$ is the more diverse skillset as each of its nine skills target different states, whereas the skills in $\pi_2$ target two different states (different skills are shown by different colored trajectories emanating from the black square agent). Thus, using the true mutual information, $I_{True}(Z; S|\pi_1) > I_{True}(Z; S|\pi_2)$. However, using the loose lower bound on mutual information employed by existing empowerment methods that penalizes differences from the current skillset, $\tilde{I}_{\text{Existing}}(Z; S|\pi_1) < \tilde{I}_{\text{Existing}}(Z; S|\pi_2)$ because $\pi_2$ is more similar to the current skillset $\pi$ than $\pi_1$. We introduce a tighter bound to the mutual information in which the more diverse skillset $\pi_1$ would score higher than $\pi_2$.

mutual information for desirable diverse skillsets that differ significantly from the current skillset. Similarly, GCRL replaces the true posterior with a fixed posterior distribution that encourages the goal-conditioned policy to execute actions that achieve the assigned goal state (Choi et al., 2021). However, in stochastic settings, this implementation can produce a loose lower bound on mutual information for diverse skillsets with abstract skills that target groupings of states, making it unlikely an agent will discover these desirable abstract skillsets. Figure 1 (Left) highlights how the way existing empowerment and GCRL methods train their variational posteriors leads to inaccurate skillset diversity measurements, and Figure 1 (Rights) illustrates the consequences of these inaccuracies.

Our main contribution, Skillset Empowerment, is an empowerment objective that maximizes a tighter lower bound on the mutual information between skills and states. In our variational lower bound on mutual information, we replace the true posterior distribution within the mutual information term for a candidate skillset with a variational posterior that is (i) conditioned on the candidate skillset and (ii) trained to match the true posterior of the candidate skillset. The resulting empowerment objective is a bandit problem, in which the actions are candidate skillsets (e.g., the parameters of the skill-conditioned policy neural network) and the reward is our version of the skillset candidate's mutual information variational lower bound, which measures the diversity of the proposed skillset. To efficiently optimize this objective despite its large action space, we introduce a new actor-critic architecture. Our experiments show that our approach can learn diverse abstract skillsets in stochastic settings, including ones with high-dimensional observations. To our knowl-

edge, our approach is the first unsupervised skill learning method to successfully learn large skillsets in stochastic settings.

## 2 BACKGROUND

### 2.1 SKILLSET MODEL AND EMPOWERMENT

We model an agent's skillset using a probabilistic graphical model defined by the tuple $(\mathcal{S}, \mathcal{A}, \mathcal{Z}, T, \phi, \pi)$. $\mathcal{S}$ is the space of states; $\mathcal{A}$ is the space of actions; $\mathcal{Z}$ is the space of skills; $T$ is the environment transition dynamics $T(s_{t+1}|s_t, a_t)$ that provides the probability of a state given the prior state and action. The transition dynamics are assumed to be conditionally independent of the history of states and actions (i.e., $T(s_{t+1}|s_t, a_t) = T(s_{t+1}|s_0, a_0, \ldots, s_t, a_t)$). The remaining distributions $\phi$ and $\pi$ are the skillset distributions that can be learned. $\phi$ represents the distribution over skills $\phi(z|s_0)$ given a skill start state $s_0$. $\pi$ represents the skill-conditioned policy $\pi(a_t|s_t, z)$ that provides the distribution over primitive actions given a state $s_t$ and skill $z$. Assuming each skill consists of $n$ primitive actions, the full joint distribution of a skill and a trajectory of actions and states $(z, a_0, s_1, \ldots, a_{n-1}, s_n)$ conditioned on a particular start state $s_0$ and skillset defined $\phi$ and $\pi$ is given by $p(z, a_0, s_1, \ldots, a_{n_1}, s_n|s_0, \phi, \pi) = \phi(z|s_0)\pi(a_0|s_0, z)p(s_1|s_0, a_0)\ldots\pi(a_{n-1}|s_{n-1}, z)p(s_n|s_{n-1}, a_{n-1})$.

In this paper, we measure the diversity of a skillset defined by $\phi$ and $\theta$ and executed from some start state $s_0$ using the mutual information between the skill random variable $Z$ and the skill-terminating state random variable $S_n$, $I(Z; S_n|s_0, \phi, \pi)$. This mutual information provides a way to measure the number of distinct skills in a skillset, in which a skill is distinct if it terminates in a set of states not targeted by other skills in the skillset. $I(Z; S_n|s_0, \phi, \pi)$ is defined

$$I(Z; S_n|s_0, \phi, \pi) = H(Z|s_0, \phi, \pi) - H(Z|s_0, \phi, \pi, S_n) \tag{1}$$

$$= \mathbb{E}_{z\sim\phi(z|s_0), s_n\sim p(s_n|s_0, \pi, z)}[\log p(z|s_0, \phi, \pi, s_n) - \log p(z|s_0, \phi)]. \tag{2}$$

Per equation 1, the diversity of a skillset grows when there are more skills in a skillset (i.e., higher skill distribution entropy $H(Z|s_0, \phi, \pi)$) and/or the skills become more distinct (i.e., the conditional entropy $H(Z|s_0, \phi, \pi, S_n)$ shrinks).

The empowerment of a state is the maximum mutual information with respect to all possible $(\phi, \pi)$ skillsets:

$$\mathcal{E}(s) = \max_{\phi, \pi} I(Z; S_n|s_0, \phi, \pi). \tag{3}$$

That is, the empowerment of a state measures the size of the most diverse skillset in that state. Note that this use of empowerment, in which mutual information is maximized to find the most diverse skillset, is different than maximizing an empowerment reward function, in which agents are encouraged to take actions that lead them to high empowerment states (Klyubin et al., 2008; Jung et al., 2012; Mohamed & Rezende, 2015). Indeed, in order to enable this other use of empowerment that is popular in the literature, there needs to be a better way to measure the empowerment of a state, which is what our work is focused on.

### 2.2 EXISTING APPROACHES TO MEASURING EMPOWERMENT

Next we discuss why both of two dominant approaches to unsupervised skill learning, empowerment-based skill learning and unsupervised GCRL, are only optimizing loose lower bounds on the mutual information between skills and states.

A key problem with optimizing the mutual information of the skill channel for a candidate skillset $(\phi, \pi)$ is that it depends on the posterior distribution $p(z|s_0, \phi, \pi, s_n)$, which provides the probability of a skill $z$ given the start state $s_0$, skillset $(\phi, \pi)$, and skill-terminating state $s_n$. As shown in line 4, the posterior is intractable to compute in large settings (e.g., domains with continuous state and action spaces) because it requires integrating over an infeasible number of trajectories.

$$p(z|s_0, \phi, \pi, s_n) = \frac{\int_{a_0, s_1, a_2, \ldots, s_{n-1}, a_{n-1}} p(z, a_0, s_1, \ldots, a_{n-1}, s_n|s_0, \phi, \pi)}{\int_{z, a_0, s_1, a_2, \ldots, s_{n-1}, a_{n-1}} p(z, a_0, s_1, \ldots, a_{n-1}, s_n|s_0, \phi, \pi)} \tag{4}$$

To overcome this problem, both existing empowerment and unsupervised GCRL methods replace the original posterior with a different variational distribution $q_\psi(z|s_0, s_n)$, similar to Mohamed & Rezende (2015) who first replaced the intractable posterior with a variational distribution when optimizing a different mutual information between open loop action sequences and terminating states. Replacing the true posterior $p(z|s_0, \phi, \pi, s_n)$ with the variational distribution $q_\psi(z|s_0, s_n)$ results in a lower bound on mutual information $\tilde{I}(Z; S_n|s_0, \phi, \pi)$ for the candidate skillset $(\phi, \pi)$. The gap between the actual mutual information $I(Z; S_n|s_0, \phi, \pi)$ and the variational lower bound on mutual information $\tilde{I}(Z; S_n|s_0, \phi, \pi)$ is an average of the KL divergences between the true posterior and the variational distribution (Barber & Agakov, 2003):

$$I(Z; S_n|s_0, \phi, \pi) - \tilde{I}(Z; S_n|s_0, \phi, \pi) = \mathbb{E}_{s_n \sim p(s_n|s_0, \phi, \pi)}[D_{KL}(p(z|s_0, \phi, \pi, s_n)||q_\psi(z|s_0, s_n))]. \tag{5}$$

Thus, replacing the true posterior with a similar variational distribution can produce a tight bound on mutual information, but replacing the true posterior with a markedly different one can produce a loose bound, which can cause the agent to significantly underestimate the diversity of a skillset. Next, we discuss the variational posteriors used by existing empowerment approaches and unsupervised GCRL, and why these can cause loose lower bounds on $\tilde{I}(Z; S_n|s_0, \phi, \pi)$.

**Existing Empowerment** The typical empowerment-based skill-learning algorithm replaces the true posterior with a variational distribution trained to be similar to the posterior of the *current* skillset $p(z|s_0, \phi_{\text{Current}}, \pi_{\text{Current}}, s_n)$ (Gregor et al., 2016; Eysenbach et al., 2018; Achiam et al., 2018; Lee et al., 2019; Choi et al., 2021; Strouse et al., 2021). Specifically, the parameters of the variational posterior $\psi$ are trained to minimize the KL divergence between the posterior of the current skillset and the variational distribution:

$$\psi^* = \underset{\psi}{\arg\min} \, \mathbb{E}_{s_n \sim p(s_n|s_0, \phi_{\text{Current}}, \pi_{\text{Current}})}[D_{KL}(p(z|s_0, \phi_{\text{Current}}, \pi_{\text{Current}}, s_n)||q_\psi(z|s_0, s_n))]. \tag{6}$$

With this posterior, the empowerment objective becomes

$$\tilde{\mathcal{E}}(s_0) = \max_{\phi(z|s_0), \pi(a|s,z)} \tilde{I}(Z; S_n|s_0, \phi, \pi),$$

$$\tilde{I}(Z; S_n|s_0, \phi, \pi) = \mathbb{E}_{z \sim \phi(z|s_0), s_n \sim p(s_n|s_0, \pi, z)}[\log q_{\psi^*}(z|s_0, s_n) - \log \phi(z|s_0)]. \tag{7}$$

Using this objective, the diversity of candidate skillsets $(\phi, \pi)$ is evaluated using the variational lower bound on mutual information defined in line 7.

The problem with this posterior implementation is that it produces a loose variational lower bound $\tilde{I}(Z; S_n|s_0, \phi, \pi)$ for desirable diverse $(\phi, \pi)$ skillsets that differ significantly from the current skillset. The loose lower bound results from the potentially large difference between the posteriors of the diverse candidate skillsets and the posterior of the current skillset. Examples of this outcome include situations similar to the one discussed in Figure 1 (Right) in which a candidate skillset can target more unique states than the current greedy skillset but because the skills $z$ that achieve the terminating states $s_n$ do not have high probability according to variational posterior $q_{\psi^*}(z|s_0, s_n)$ at the states $s_n$, the $\log q_{\psi^*}(z|s_0, s_n)$ can be very low, which can then result in a low diversity score $\tilde{I}(Z; S_n)$ for the candidate skillset $(\phi, \pi)$ that significantly underestimates its true diversity. The low $\tilde{I}(Z; S_n)$ scores would then discourage the agent from selecting these skillsets. Instead, the skillsets that are favored are the ones similar to the current skillset because the $(z, s_n)$ tuples generated by skillsets similar to the current skillset will be rewarded with higher $\log q_{\psi^*}(z|s_0, s_n)$ values. Several prior works have empirically demonstrated this result in which existing empowerment approaches tend to learn skillsets that do not change much from initialization (Campos et al., 2020; Park et al., 2022; 2023a;b; Strouse et al., 2021; Levy et al., 2023).

**Unsupervised GCRL** Unsupervised GCRL approaches replace the true posterior with a fixed variational posterior that encourages the skill-conditioned policy (or the goal-conditioned policy) to target the conditioned goal state. There are different options for the fixed variational posterior depending on the desired goal-conditioned reward. The variational posterior could take the form of a fixed standard deviation gaussian centered at the skill-terminating state $s_n$: $q(z|s_0, s_n) = \mathcal{N}(z; \mu = s_n, \sigma = \sigma_0)$ (Choi et al., 2021). This will encourage the learning of a goal-conditioned policy such that when given a goal state $z$, the goal-conditioned policy will target a skill-terminating state $s_n$ close to $z$.

For the desirable diverse skillsets, the tightness of a variational lower bound on mutual information $\tilde{I}(Z; S_n|s_0, \phi, \pi)$ that employs a fixed posterior can depend on the level of randomness in the domain. In deterministic settings, the variational lower bound $\tilde{I}(Z; S_n|s_0, \phi, \pi)$ for diverse skillsets can form a tight bound to the true mutual information $I(Z; S_n|s_0, \phi, \pi)$. Consider the $\tilde{I}$ of an effective goal-conditioned policy $\pi$, in which the fixed gaussian posterior $q(z|s_n)$ takes the form of a tight fixed-variance gaussian with a mean at the skill-terminating state $s_n$. Because the goal-conditioned policy is effective (i.e., for goal state $z$, $s_n \approx z$), the true posterior $p(z|s_0, \phi, \theta, s_n)$ will be similar to the fixed-variance gaussian variational posterior, and thus the gap between the true mutual information and $\tilde{I}$ will be small. In deterministic settings, GCRL can thus be an effective way to learn diverse skillsets, which has been repeatedly demonstrated empirically (Andrychowicz et al., 2017; Mendonca et al., 2021; Nair et al., 2018; Pong et al., 2019; Campos et al., 2020; Pitis et al., 2020; Held et al., 2017; McClinton et al., 2021; Held et al., 2017; Kim et al., 2023; Levy et al., 2019).

However, in significantly stochastic settings in which specific states cannot be consistently achieved, the lower bound can be quite loose for the most diverse skillsets. In significantly random domains, the most diverse skillsets may be ones that contain abstract skills that target distinct groupings of states. Abstract skillsets like these will have posteriors in which many skill-terminating states map to the same skill, which can be far different than the fixed posterior used in GCRL in which each terminating state is mapped to its own unique goal state. Thus, the GCRL lower bound on mutual information can be quite loose for these desirable abstract skillsets, which will discourage the agent from learning them. Instead, the GCRL objective may limit skillsets to the more deterministic parts of the environment because trying to achieve any goal state that cannot be reliably achieved may be heavily penalized.

## 3 SKILLSET EMPOWERMENT

To enable agents to learn diverse skillsets in stochastic domains, we introduce the empowerment objective, Skillset Empowerment. In this section, we first present the objective and explain how the objective measures a tighter lower bound on the empowerment of a states, meaning that our approach can learn larger sets of skill in a state relative to existing approaches.

### 3.1 SKILLSET EMPOWERMENT OBJECTIVE

The Skillset Empowerment objective is defined as follows:

$$\mathcal{E}^{SE}(s_0) = \max_{\phi(z|s_0), \pi(a|s,z)} \tilde{I}^{SE}(Z; S_n|s_0, \phi, \pi),$$

$$\tilde{I}^{SE}(Z; S_n|s_0, \phi, \pi) = \mathbb{E}_{z \sim \phi(z|s_0), s_n \sim p(s_n|s_0, \pi, z)}[\log q_{\psi^*}(z|s_0, \phi, \pi, s_n) - \log \phi(z|s_0)], \quad (8)$$

$$\psi^* = \underset{\psi}{\operatorname{argmin}} \, \mathbb{E}_{s_n \sim p(s_n|s_0, \phi, \pi)}[D_{KL}(p(z|s_0, \phi, \pi, s_n)||q_\psi(z|s_0, \phi, \pi, s_n))]$$

Like existing approaches, Skillset Empowerment optimizes a variational lower bound on the mutual information between skills and states in which the true posterior of a candidate skillset $(\phi, \pi)$ is replaced with a variational distribution. The key reason Skillset Empowerment optimizes a tighter bound on mutual information and thereby measures a tighter bound on the empowerment of a state is the manner in which $\psi^*$ is selected. For any candidate skillset $(\phi, \pi)$, Skillset Empowerment trains the variational posterior $q_\psi(z|s_0, \phi, \pi, s_n)$ to match the true posterior of the candidate skillset $p(z|s_0, \phi, \pi, s_n)$. Note that the variational posterior is now additionally conditioned on the skillset distributions $(\phi, \pi)$ in contrast to existing empowerment approaches. The next section will discuss how this is implemented in practice. This strategy for learning $\psi^*$ results in a variational mutual information $\tilde{I}^{SE}(Z; S_n|s_0, \phi, \pi)$ that is a tighter lower bound on the true mutual information $I(Z; S_n|s_0, \phi, \pi)$ for any $(\phi, \pi)$ skillset because the KL divergence between the true and variational posterior cannot be larger than the KL divergence in existing approaches. This is true because in the selection of $\psi^*$, Skillset Empowerment can choose the $\psi$ selected by existing empowerment approaches or a $\psi$ that produces a fixed variational distribution like in unsupervised GCRL if that is the $\psi$ that minimizes the KL divergence, but Skillset Empowerment is not limited to those options. The tighter bound on the mutual information between skills and states for any $(\phi, \pi)$ skillset means that Skillset Empowerment also measures a tighter bound on the empowerment of a state (see section A of the appendix for proof).

## 3.2 PRACTICAL IMPLEMENTATION

Next, we provide a practical implementation that optimizes the objective with deep learning. We first discuss how we optimize the skillset distributions $\phi$ and $\pi$ because that informs how we will train the variational parameters $\psi$. Before discussing how the skillset distributions are updated updated, note that the Skillset Empowerment objective is not a typical skill-conditioned Reinforcement Learning (RL) problem in which rewards are only functions of states, skills, and actions. If Skillset Empowerment were to be converted to an RL problem, the reward function for the final action would be $\log q_\psi(z|s_0, \phi, \pi, s_n)$, which depends in part on the skill-conditioned policy $\pi$. Optimizing Skillset Empowerment with a typical skill-conditioned RL approach, in which the agent learns a policy mapping states and skills to primitive actions, would be subject to potentially significant non-stationary rewards because each time the skill-conditioned policy is updated, the final step reward could change.

Instead we optimize the Skillset Empowerment with respect to the skillset distributions as a bandit problem, in which actions are $(\phi, \pi)$ skillsets and rewards are the diversity of those skillsets as measured by $\tilde{I}^{SE}(Z; S_n|s_0, \phi, \pi)$. In order for the skillset distributions $\phi$ and $\pi$ to serve as actions, we first describe how Skillset Empowerment vectorizes these distributions. In our implementation of Skillset Empowerment, we represent the skill-conditioned policy $\pi$ as a vector containing the the weights and biases of a neural network $f_\pi$ that forms the skill-conditioned policy. $f_\pi : \mathcal{S} \times \mathcal{Z} \rightarrow \mathcal{A}$ takes as input a state $s$ and skill $z$ outputs the mean of a gaussian skill-conditioned policy. That is, $\pi(a|s, z) = \mathcal{N}(a; \mu = f_\pi(s, z), \sigma = \sigma_0)$, in which $\sigma_0$ is a small fixed standard deviation. We represent $\phi(z|s_0)$ using a scalar that represents a uniform distribution that takes the shape of a $d$-dimensional cube with side length $\phi$. For instance, in our tasks in which the skills are two-dimensional, skills are sampled from a square with side length $\phi$ centered at the origin. Figure 3 in the Appendix provides an illustration of a two-dimensional skill distribution $\phi$. The probability density $\phi(z|s_0) = (1/\phi)^d$.

We optimize both $\phi$ and $\pi$ using their own bandit problem. In the $\phi$ bandit problem, the bandit policy is $f_\mu : \mathcal{S} \rightarrow \phi$, which takes as input the skill start state $s_0$ and outputs $\phi$ (i.e., a scalar representing the size of the uniform skill space). In the $\pi$ bandit problem, the bandit policy $f_\lambda : \mathcal{S} \times \phi \rightarrow \pi$ takes as input the skill start state $s_0$ and a $\phi$ value and outputs $\pi$, the vector of parameters that define the skill-conditioned policy. In the $\pi$ bandit problem, the reward for an $\pi$ action is the Skillset Empowerment variational lower bound on mutual information $\tilde{I}^{SE}(Z; S_n|s_0, \phi, \pi)$ (equation 8) (i.e., the reward for proposing the skillset $(\phi, \pi)$ is is the diversity of the skillset). Similarly, the reward in the $\phi$ bandit problem for a $\phi$ action is $\tilde{I}^{SE}(Z; S_n|s_0, \phi, \pi = f_\lambda(s_0, \phi))$, in which the skill-conditioned policy $\pi$ is the greedy output from the $\pi$ bandit policy. Both bandit policies are optimized using an actor-critic structure. That is, to determine the gradient for the bandit policy $f_\mu$, a critic $Q_\rho(s_0, \phi)$ is trained to approximate $\tilde{I}^{SE}(Z; S_n)(s_0, \phi, f_\lambda(s_0, \phi))$ for $\phi$ values around the current greedy output $f_\eta(s_0)$. Similarly, a critic $Q_\omega(s_0, \phi, \pi)$ is used to approximate $\tilde{I}^{SE}$ for sets of parameters $\pi$ that are near the current greedy vector $f_\lambda(s_0, \phi)$. Figure 4 in the Appendix provides a visual overview of the two bandit problems.

The current setup is not yet practical because trying to learn the critic $Q_\omega(s_0, \phi, \pi)$ for the $\pi$ actor when the vector of parameters $\pi$ may be thousands of dimensions is infeasible. $Q_\omega$ would need to be able to discern the difference in mutual information when small changes are made to numerous parameters in $\pi$. Instead, because the gradient with respect to the $\pi$ bandit policy $f_\lambda$ only needs to know how the $\tilde{I}^{SE}$ reward responds to small changes in each of the individual parameters in $\pi$, we instead train $|\pi|$ critics, $Q_{\omega_0}(s_0, \phi, \pi_0), \ldots, Q_{\omega_{|\pi|-1}}(s_0, \phi, \pi_{|\pi|-1})$. Each of these critics only takes a scalar, $\pi_i$, as input, in which $\pi_i$ represents the $i$-th parameter of $\pi$ (e.g., could be a noisy value of the current greedy $i$-th parameter $f_\lambda(s_0, \phi)[i]$). The remaining parameters of $\pi$ are assumed to take on their current greedy values from $f_\lambda(s_0, \phi)$. Thus, each critic $Q_i(s_0, \phi, \pi_i)$ only needs to approximate how the mutual information changes from small changes to a weight or bias parameter in $\pi$. All of these critics are updated in parallel. In Figure 5 in the Appendix we show how each of these $|\pi|$ critics attach to the bandit policy $f_\lambda$ that outputs $\pi$.

Prior to updating the critics for each parameter of the skill-conditioned policy $\pi$, the variational distribution parameters $\psi$ need to be updated so that $\tilde{I}^{SE}(Z; S_n|s_0, \phi, \pi)$ forms a tighter bound on the true mutual information $I(Z; S_n|s_0, \phi, \pi)$ (i.e., more accurately measures the diversity of

the $(\phi, \pi)$ skillset). Because we need to estimate $\tilde{I}^{SE}$ for small changes in each of the parameters of $\pi$, we train $|\pi|$ different sets of variational parameters $\psi_0, \ldots, \psi_{|\pi|-1}$. Each set of variational parameters $\psi_i$ is trained in parallel to minimize the KL divergence between $p(z|s_0, \phi, \pi_i, s_n)$ and $q_{\psi_i}(z|s_0, \phi, \pi_i, s_n)$.

---

**Algorithm 1** Skillset Empowerment

---
   **repeat**
       Update skill-conditioned policy critics $Q_{\omega_i}$ (see Equation 17)
       Update skill-conditioned policy actor $f_\lambda$ (see equation 18)
       Update skill distribution critic $Q_\rho$ (see equation 19)
       Update skill distribution actor $f_\mu$ (see equation 20)
   **until** convergence

---

Section B in the Appendix provides the specific objective functions for updating the parameter-specific critics, skill-conditioned policy actor, skill distribution critic, and skill distribution actor. Algorithm 1 provides the full algorithm for the practical implementation of Skillset Empowerment. Section C of the Appendix discusses how the $|\pi|$ sets of variational posteriors and the $|\pi|$ skill-conditioned policy critics can be training efficiently in parallel with the help of multiple accelerators and the parallelization capabilities of modern deep learning frameworks (e.g., JAX).

### 3.3 LIMITATIONS

The major limitation of Skillset Empowerment is that it requires a model of the transition dynamics $p(s_{t+1}|s_t, a_t)$ (i.e., access to simulator of the environment) in order to generate the large number of (skill $z$, skill-terminating state $s_n$) tuples needed to train all of the parameter-specific critics. However, subsequent work by Author (2024) (included as an anonymous supplementary file), which builds on Skillset Empowerment, shows how large skillsets can be learned without a simulator. Instead of maximizing the mutual information between skills and states, Author (2024) maximize a different objective that has the same optimal skillset as $I(Z; S_n)$ but only requires learning a latent-predictive model, which is significantly easier to learn than a simulator as it operates in a lower-dimensional latent space.

## 4 EXPERIMENTS

### 4.1 ENVIRONMENTS

We apply Skillset Empowerment and a pair of baselines to several domains. In terms of stochasticity and the dimensionality of observations, most of these environments are complex because they have highly stochastic transition dynamics and some include high-dimensional state observations. On the other hand, in terms of the dimensionality of the underlying state space not visible by the agent, most domains have simple, low-dimensional underlying state spaces. Stochastic domains are used because general purpose agents need to be able to build large skillsets in environments with significant randomness, and there are already effective algorithms for learning skills in deterministic settings such as unsupervised goal-conditioned RL methods. Low-dimensional underlying state environments are used in order to limit the parallel compute needed to train Skillset Empowerment agents because larger skill spaces require more $(z, s_n)$ data to train the parameter-specific critics.

The first two experiments are built in a stochastic four rooms setting. In the navigation version of this setting, a 2D point agent executes 2D (i.e., $(\Delta x, \Delta y)$) actions in a setting with four separated rooms. After each action is complete, the agent is moved randomly to the corresponding point in one of the four rooms. In the pick-and-place version of this setting, there is a two-dimensional object the agent can move around if the agent is within a certain distance. The abstract skills agents should learn in these domains are to target $(x, y)$ offset positions from the center of a room for the agent (and for the object in the pick-and-place version). The other two stochastic environments are built in a RGB-colored QR code domain, in which a 2D agent moves within a lightly-colored QR code where every pixel of the QR code changes after each action. The state observations are 432 dimensional (12x12x3 images). We also created a pick-and-place version of this task, in which the agent can move around an object provided the object is within reach. The abstract skills to learn in these

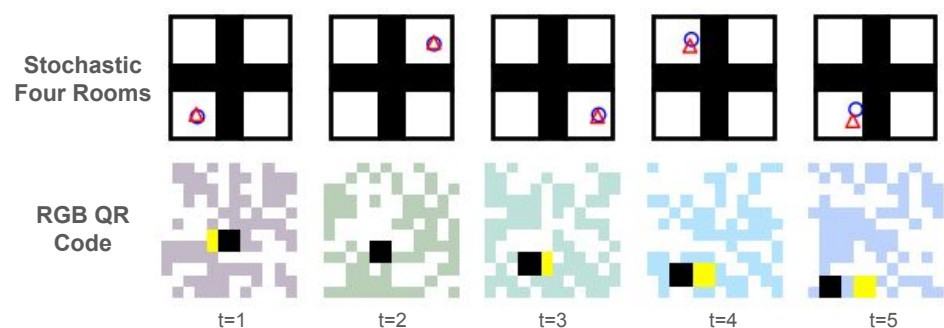

Figure 2: Sample skill sequences in the pick-and-place versions of the Stochastic Four Rooms and RGB QR Code domains. In top row, the blue circle agent executes a skill to carry the red triangle object to the right side of a room. In bottom row, the black square agent carries the yellow object to bottom of room.

domains are again to target $(x, y)$ locations for the agent (and object in the pick-and-place version). Image sequences showing executed skills in the pick-and-place versions of the stochastic tasks are shown in Figure 2. In addition to the stochastic domains, we also applied Skillset Empowerment to the Continuous Mountain Car domain (Towers et al., 2024) and the Ant domain in Brax (Freeman et al., 2021) to test whether agents can learn skills to target states containing both positions and velocities. We also tested the Ant domain to see how the algorithm would perform in a domain with a much larger underlying state space (29-dim). Additional details for these domains are provided in section D of the Appendix.

Given our purpose of using empowerment to learn large skillsets, we evaluate the performance of Skillset Empowerment and the baselines by the size of the skillsets they learn in each domain. We measure the size of the skillsets using the variational mutual information $\tilde{I}(Z; S_n | s_0, \phi, \pi)$ from a single start state $s_0$. In this paper, we are not assessing performance on downstream tasks, in which, for instance, a hierarchical agent needs to learn a higher level policy that executes skills from the learned skillsets to maximize some reward function. However, in section E of the Appendix we describe how it is simple to implement such hierarchical agents that use the Skillset Empowerment $(\phi, \pi)$ skillsets as a temporally extended action space.

### 4.2 BASELINES

We compare our approach, Skillset Empowerment, to both an empowerment-based skill-learning method and unsupervised GCRL. For the prior empowerment-based method, we selected Variational Intrinsic Control (VIC) (Gregor et al., 2016). VIC, like other approaches to measuring empowerment including DIAYN (Eysenbach et al., 2018) and VALOR (Achiam et al., 2018), maximizes a loose lower bound on the mutual information between skills and states because the variational posterior is trained to match the posterior of the current greedy skillset and not the candidate skillset. For the unsupervised GCRL comparison, our focus was solely on whether goal-conditioned skills can be learned in stochastic domains and not on exploration which is the primary focus of recent unsupervised GCRL algorithms. Thus, we assist the unsupervised GCRL algorithm and provide it with the distribution of reachable states to use as the distribution of goal states and thereby are just comparing our algorithm to supervised GCRL. We compare to a GCRL objective in which the variational posterior used is the tight diagonal gaussian centered at the skill-terminating state discussed in section 2.2. For the higher dimensional QR code tasks, we implemented Reinforcement learning with Imagined Goals (RIG) (Nair et al., 2018). RIG performs GCRL in a latent space learned separately by a VAE. See section F for details on how the baselines were implemented.

### 4.3 RESULTS

Table 1 shows the size of the skillsets learned by all algorithms in the stochastic settings. Skillset size is measured with the variational mutual information $\tilde{I}(Z; S_n)$. Note that mutual information in measured on a logarithmic scale (in this case, nats) so the 8.7 nats of skills learned by Skillset Em-

Table 1: Average learned skillset sizes for all baselines as measured by variational empowerment $\tilde{I}(Z; S_n)$. Average and standard deviation computed from 5 random seeds.

| Task | Ours | VIC | GCRL |
|------|------|-----|------|
| Four Rooms Nav. | **5.1± 0.3** | 0.2± 0.4 | 0.3± 0.4 |
| Four Room P.-and-P. | **8.7± 0.3** | -0.1± 0.3 | 3.9± 0.6 |
| RGB QR Nav. | **3.5± 0.1** | -0.4± 0.0 | -0.4± 0.3 |
| RGB QR P.-and-P. | **6.0± 0.2** | -0.6± 0.1 | -2.6± 5.8 |

powerment in the pick-and-place version of the Stochastic Four Rooms domain means that Skillset Empowerment learned $e^{8.7} \approx 6,000$ skills. In the Continuous Mountain Car and Ant domains, Skillset Empowerment learned skillsets with sizes of $5.3 \pm 0.3$ and $16.8 \pm 1.8$ nats, respectively. The results show that Skillset Empowerment was able to learn large skillsets in all domains, while both VIC and GCRL were unable to learn meaningful skillsets in the stochastic settings. Note that because skillset sizes were measured with variational mutual information, which is a lower bound on the mutual information between skills and states, skillset size results can be negative. This happens with training goes poorly and the true posterior distribution $p(z|s_n)$ of the learned skillset is significantly different from the diagonal gaussian variational posterior $q_\psi(z|s_n)$.

For additional evidence that Skillset Empowerment is able to learn large skills in all domains, we provide visualizations of the mutual information entropy terms (i.e., $H(S_n), H(S_n|Z), H(Z), H(Z|S_n)$) both before and after training in Figures 6-16 in the Appendix. The $H(S_n)$ visuals show the skill-terminating states $s_n$ achieved by 1000 skills randomly sampled from the learned skill distribution. In all the stochastic settings and Continuous Mountain Car, the skill-terminating states nearly uniformly cover the reachable state space. To show that this was not achieved by simply executing a policy that uniformly samples actions from the action space, in the center image we visualize $H(S_n|Z)$, which shows 12 skill-terminating states $s_n$ from four randomly selected skills from the skill distribution. In the stochastic settings, for instance, the $s_n$ generated by each skill $z$ target a specific $(x, y)$ offset location for the agent and an $(x, y)$ offset location for the object in the pick-and-place tasks, which is the correct abstract skill to learn. These visuals also visualize $H(Z)$ by showing the distribution over skills $\phi$ that takes the shape of a $d$-dimensional cube. Lastly, we visualize $H(Z|S_n)$ by showing four randomly selected skills $z$ and samples from the learned posterior $q_\psi(z|s_0, \phi, \pi, s_n)$. As expected for a diverse skillset in which different skills target different states, these samples of the posterior distribution tightly surround the original skill. In addition, Figure 16 shows entropy visualizations for the Ant domain. In this setting, Skillset Empowerment learned a very large skillset of 16.8 nats ($\approx 20$ million skills). However, as shown by the visual of $H(S_n^{x,y})$ showing the final torso locations, most of these skills are small rotations of the body or bending of the joints and not translations of the torso. We believe this is mostly a result of being compute constrained. We would need to sample much larger batches of $(z, s_n)$ tuples to properly measure the diversity of skillsets of this size, but this would require more compute.

We note that searching across the space of $(\phi, \pi)$ skillsets for a skillset that targets a diverse distribution of skill-terminating states is not a trivial task in these domains. A skill-conditioned policy that randomly executes actions would produce a zero mutual information skillset. A skillset that tried to maximize the mutual information between skills and open loop action sequences $I(Z; A)$ (i.e., have each skill execute a different action) would also produce relatively low $I(Z; S_n)$ because among the space of open loop action sequences $a_0, a_1, \ldots, a_{n-1}$, many of these sequences target the same skill-terminating state $s_n$. In addition, the need to have the skillset fit a diagonal gaussian variational posterior $q_\psi(z|s_0, \phi, \pi, s_n)$ also makes the task challenging because a skillset in which distant skills $z$ target the same state $s_n$ can produce a low $\tilde{I}(Z; S_n)$ score because this would result in a high entropy variational posterior $q_\psi$. Instead, each small region of the skill distribution needs to target a distinct grouping of states $s_n$.

Moreover, the baselines were not able to learn large skillsets in any of the settings. For instance, in the stochastic four rooms task, the GCRL agent only learns skills to move to corners of the room as shown in Figure 17, which shows the skill-terminating states of 1000 random skills. More specifically, as shown in Figure 18, when given a goal state of some (x,y) position in one of the four rooms, the agent simply moves towards whichever room the goal is in regardless of where in

the room the goal is. This behavior is likely taken to minimize the average distance to the goal as the goal-conditioned reward heavily penalizes the agent if it is far from the specific goal state. In the image-based QR code tasks, the VAE generally struggled to reconstruct the large variety of colored QR codes as shown in Figure 19, ultimately producing an image similar to a mean QR code. The overly abstract latent state space in turn made it challenging for the GCRL component to learn distinct skills. These results provide evidence that GCRL's loose lower bound on mutual information for diverse abstract skillsets does discourage agents from learning these desirable skillsets. On the other hand, because Skillset Empowerment can learn a tight bound on mutual information for diverse abstract skillsets, agents are encouraged to learn them. In addition, like GCRL, the performance of VIC agents also was poor. As with prior works, we observed stagnant skillsets.

## 5 RELATED WORK

There have been many prior works that have tried to use empowerment to learn large skillsets. Early empowerment methods showed how mutual information between actions and states could be optimized in small settings with discrete state and/or action spaces (Klyubin et al., 2008; Salge et al., 2013a; Jung et al., 2012). Several later works integrated variational inference techniques that enable empowerment-based skill learning to be applied to larger continuous domains (Mohamed & Rezende, 2015; Karl et al., 2017; Gregor et al., 2016; Eysenbach et al., 2018; Sharma et al., 2019; Li et al., 2019; Hansen et al., 2020). However, these methods were limited in the size of skillsets they were able to learn as they only maximize a loose lower bound on mutual information, making it difficult to accurately measure the diversity of a skillset.

Complementary to our work are the methods that use empowerment for downstream applications, such as using empowerment as a state utility function (Klyubin et al., 2008; Salge et al., 2013b; Jung et al., 2012; Mohamed & Rezende, 2015; Karl et al., 2015; 2017), as a temporally extended action space for hierarchical RL (Eysenbach et al., 2018; Levy et al., 2023), as an evolutionary signal to evolve sensors and actuators (Klyubin et al., 2005), as an objective for learning a state representation (Capdepuy, 2011; Bharadhwaj et al., 2022), as an intrinsic motivation reward (Oudeyer & Kaplan, 2007; Bharadhwaj et al., 2022), and as a way to measure human empowerment (Du et al., 2020; Myers et al., 2024).

Also related to our work are the methods that learn abstract skills in settings with random environment distractors (Bharadhwaj et al., 2022; Fu et al., 2021; Ma et al., 2021; Zhang et al., 2020; Rudolph et al., 2024; Zou & Suzuki, 2024). However, these approaches only learn a single skill with the help of a reward function or require supervision in the form of a hand-crafted goal space. To our knowledge, our approach is the first unsupervised skill learning method to successfully learn large skillsets in stochastic settings.

## 6 CONCLUSION

Agents need to be able to execute large skillsets in settings with significant randomness and empowerment should be able to help. We show a major reason why previous empowerment methods have been unable to learn large skillsets is that they have been maximizing a loose lower bound on the mutual information between skills and states. To overcome this problem, we introduced the Skillset Empowerment algorithm, which maximizes a tighter bound on the mutual information between skills and states using a new actor-critic architecture. We show empirically that Skillset Empowerment enables agents to learn large skillsets in a variety of settings, includes ones with stochastic transition dynamics and high-dimensional state observations.

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

# A  SKILLSET EMPOWERMENT'S TIGHTER BOUND ON EMPOWERMENT PROOF

$$\mathcal{E}^{\text{Existing}}(s_0) = \tilde{I}^{\text{Existing}}(Z; O_n | s_0, \phi^*_{\text{Existing}}, \pi^*_{\text{Existing}}) \tag{9}$$

$$\leq \tilde{I}^{SE}(Z; O_n | s_0, \phi^*_{\text{Existing}}, \pi^*_{\text{Existing}}) \tag{10}$$

$$\leq \tilde{I}^{SE}(Z; O_n | s_0, \phi^*_{SE}, \pi^*_{SE}) \tag{11}$$

$$= \mathcal{E}^{SE}(s_0) \tag{12}$$

$$\leq I(Z; O_n | s_0, \phi^*_{SE}, \pi^*_{SE}) \tag{13}$$

$$\leq I(Z; O_n | s_0, \phi^*, \pi^*) \tag{14}$$

$$= \mathcal{E}(s_0). \tag{15}$$

Line 9 provides the variational empowerment learned by existing approaches, in which $(\phi^*_{\text{Existing}}, \pi^*_{\text{Existing}})$ are the mutual information maximizing distributions learned by the existing approach. As shown in Line 10, this is less than or equal to the mutual information learned by Skillset Empowerment using the same skillset distributions $(\phi^*_{\text{Existing}}, \pi^*_{\text{Existing}})$. This is true because the gap between the true mutual information $I(Z; O_n|s_0, \phi^*_{\text{Existing}}, \pi^*_{\text{Existing}})$ and the variational mutual information found by existing approaches $\tilde{I}^{\text{Existing}}(Z; O_n|s_0, \phi^*_{\text{Existing}}, \pi^*_{\text{Existing}})$ cannot be smaller than the gap between the true mutual information and the variational mutual information learned by Skillset Empowerment, $\tilde{I}^{SE}(Z; O_n|s_0, \phi^*_{\text{Existing}}, \pi^*_{\text{Existing}})$. The gap cannot be smaller because the gap depends on the KL divergence between the true and variational posteriors and Skillset Empowerment can always choose the same variational parameters $\psi$ as existing approaches. Subtracting these two differences produces the inequality, $\tilde{I}^{SE}(Z; O_n|s_0, \phi^*_{\text{Existing}}, \pi^*_{\text{Existing}}) \geq \tilde{I}^{\text{Existing}}(Z; O_n|s_0, \phi^*_{\text{Existing}}, \pi^*_{\text{Existing}})$. In line 11, the inequality results because the skillset distributions $(\phi^*_{SE}, \pi^*_{SE})$ that maximize the Skillset Empowerment version of variational mutual information are selected. Line 12 uses the definition of Skillset Empowerment. The inequality in line 13 results because of the gap between the variational mutual information learned by Skillset Empowerment and the true mutual information $I(Z; O_n|s_0, \phi^*_{SE}, \pi^*_{SE})$. The inequality in line 14 because $(\phi^*_{SE}, \pi^*_{SE})$ may not be the maximizing skillset distributions $(\phi^*, \pi^*)$. The last line uses the definition of empowerment

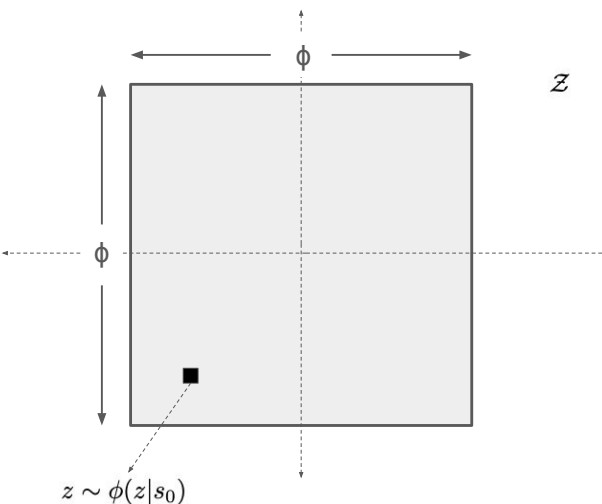

Figure 3: Illustration of the uniform distribution over skills $\phi$ used by Skillset Empowerment. The uniform distribution takes the shape of a $d$-dimensional cube centered at the origin with side length $\phi$. For instance, if the dimensionality of the skill space is 2 (i.e., $d = 2$) as in the figure, skills $z \sim \phi(z|s_0)$ are uniformly sampled from a square centered at the origin with side length $\phi$.

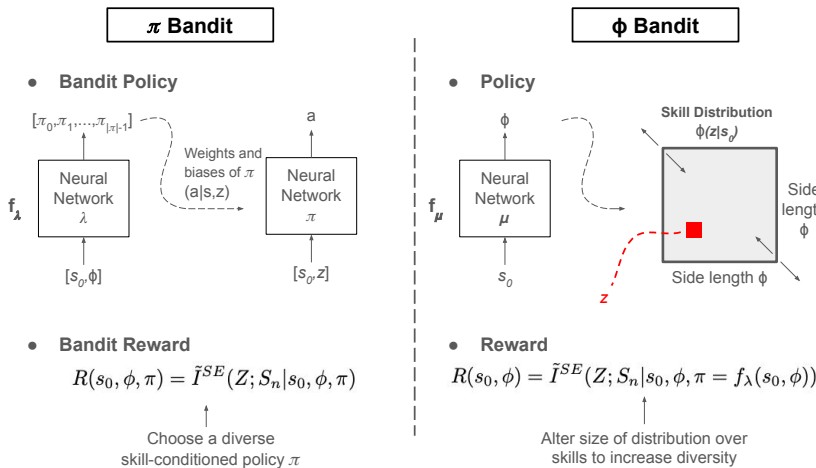

Figure 4: Overview of the two bandit problems. In the $\pi$ bandit problem, the task is to optimize the bandit policy $f_\lambda : \mathcal{S} \times \phi \rightarrow \pi$, which takes as input the skill start state $s_0$ and side length of the uniform distribution $\phi$ and output a vector $\pi$ consisting of the parameters of the skill-conditioned policy neural network. That is, $\pi$ is the vector of the weights and biases of the function $f_\pi$ that determines of the mean of an action given a state and skill. The reward for a $\pi$ action $R(s_0, \phi, \pi)$ is the mutual information lower bound $\tilde{I}^{SE}(Z; S_n | s_0, \phi, \pi)$ (i.e., the reward is how diverse the skillset $(\phi, \pi)$ is). In the $\phi$ bandit problem, the task is to learn the bandit policy $f_\mu : \mathcal{S} \rightarrow \phi$, which takes as input the skill start state $s_0$ and outputs $\phi$, which is the side length of the $d$-dimensional cube uniform distribution centered at the origin. The reward for a particular $\phi$ action $R(s_0, \phi)$ is $\tilde{I}^{SE}(Z; S_n | s_0, \phi, \pi = f_\lambda(s_0, \phi))$. That is, the bandit policy is encouraged to output $\phi$ that results in diverse skillsets $(\phi, \pi = f_\lambda(s_0, \phi))$.

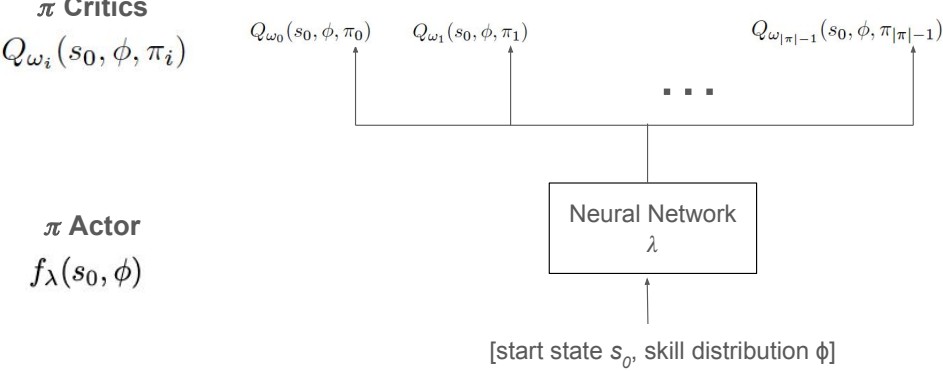

Figure 5: Illustration of how the parameter-specific critics, $Q_{\omega_i}$ for $i = 0. \ldots, |\pi| - 1$, attach to the actor $f_\lambda$ in order to determine the gradients of the actor. For each parameter $i$ in $\pi$, a critic $Q_{\omega_i}$ approximates how the diversity of the skill-conditioned policy changes with small changes to the $i$-th parameter of $\pi$. To obtain gradients showing how the diversity of a skill-conditioned policy changes with respect to $\lambda$, gradients are thus passed through each of the parameter-specific critics.

## B  ACTOR-CRITIC OBJECTIVE FUNCTIONS

### B.1  SKILL-CONDITIONED POLICY $\pi$ ACTOR-CRITIC

To update the parameter-specific critics, $Q_{\omega_0}, \ldots, Q_{\omega_{|\pi|-1}}$, so that they accurately measure the diversity of $(\phi, \pi_i)$ skillsets, a two step process is followed. In the first step, the parameter-specific variational parameters $\psi_0, \ldots, \psi_{|\pi|-1}$ are updated by maximizing the following maximum likelihood objective with respect to the variational posterior parameters $\psi_i$ for all $i = 0, \ldots, |\pi| - 1$ in

parallel:

$$J(\psi_i) = \mathbb{E}_{s_0 \sim p(s_0), \phi \sim \hat{f}_\mu, \pi_i \sim \hat{f}_\lambda}[\tilde{I}^{SE}(Z; S_n|s_0, \phi, \pi)] \tag{16}$$

$$= \mathbb{E}_{s_0 \sim p(s_0), \phi \sim \hat{f}_\mu, \pi_i \sim \hat{f}_\lambda, z \sim \phi(z|s_0), s_n \sim p(s_n|s_0, \pi_i, z)}[\log q_\psi(z|s_0, \phi, \pi_i, s_n)].$$

In the outer expectation, the skill start state $s_0$ is sampled from the skill start state distribution $p(s_0)$; $\phi$ is sampled by adding noise to the greedy output of the skill distribution actor $f_\mu(s_0)$; $\pi_i$, which represents the $i$-th parameter of the skill-conditioned policy, is sampled by adding noise to the $i$-th parameter of $f_\lambda(s_0, \phi)$.

In the second step, the parameter-specific critics are updated so that they better approximate the variational mutual information $\tilde{I}(Z; S_n|s_0, \phi, \pi_i)$ using the updated variational posterior parameters. This is done by minimizing the follow supervised learning objective for all critic parameters $\omega_0, \ldots, \omega_{|\pi|-1}$ in parallel.

$$J(\omega_i) = \mathbb{E}_{s_0 \sim p(s_0), \phi \sim \hat{f}_\mu, \pi_i \sim \hat{f}_\lambda}[(Q_{\omega_i}(s_0, \phi, \pi_i) - \text{Target}(s_0, \phi, \pi_i))^2], \tag{17}$$

$$\text{Target}(s_0, \phi, \pi_i) = \mathbb{E}_{z \sim \phi(z|s_0), s_n \sim p(s_n|s_0, \pi_i, z)}[\log q_\psi(z|s_0, \phi, \pi_i, s_n)].$$

The skill-conditioned policy actor $f_\lambda$ is then updated by maximizing the following objective function with respect to $\lambda$.

$$J(\lambda) = \mathbb{E}_{s_0 \sim p(s_0), \phi \sim \hat{f}_\mu}\left[\sum_{i=1}^{|\pi|-1} Q_{\omega_i}(s_0, \phi, f_\lambda(s_0, \phi)[i])\right], \tag{18}$$

in which $f_\lambda(s_0, \phi)[i]$ outputs the $i$-th parameter of the skill-conditioned policy actor. Thus, all the parameter-specific critics are used to obtain the gradient with respect to the skill-conditioned policy actor parameters $\lambda$.

## B.2 Skill Distribution $\phi$ Actor-Critic

The skill distribution critic $Q_\rho(s_0, \phi)$ is updated by minimizing the following supervised learning objective with respect to $\rho$:

$$J(\rho) = \mathbb{E}_{s_0 \sim p(s_0), \phi \sim f_\mu(s_0)}[(Q_\rho(s_0, \phi) - Q_{\omega_0}(s_0, \phi, f_\lambda(s_0, \phi)))^2]. \tag{19}$$

That is, the target Q value for the skill distribution critic is the skill-conditioned policy critic value $Q_{\omega_0}(s_0, \phi, f_\lambda(s_0, \phi))$.

The skill distribution actor $f_\mu$ is updated by maximizing the following objective with respect to $\mu$:

$$J(\mu) = \mathbb{E}_{s_0 \sim p(s_0)}[Q_\rho(s_0, f_\mu(s_0))]. \tag{20}$$

## C  Parallelizing Training

Although the number of skill-conditioned policy parameters may be large, the $|\pi|$ sets of variational posterior and $\pi$ critic parameters can be trained efficiently using the parallelization capabilities of modern deep learning frameworks (e.g., JAX) and multiple accelerators. For instance, the update step for each set variational posterior parameters $\psi_i$ requires (skill $z$, state $s_n$) tuples from skillsets $(\phi, \pi_i)$. Assuming access to multiple accelerators, this update step can occur in parallel for all $|\pi|$ sets of variational posterior parameters using the pmap and vmap functions in JAX. For instance, if the skill-conditioned policy $\pi$ has 1000 dimensions and there are 4 GPUs available, each GPU can process the updates to 250 sets of variational posterior parameters in parallel.

Table 2 provides some data on the parallel computation demands of our approach for each of our experiments. $|\pi|$ is the number of parameters in the skill-conditioned policy. Update Time reflects the time (in seconds) required to complete one whole update step (i.e., one iteration of the Repeat loop in Algorithm 1). Note that the update times shown for the four rooms tasks were when using a single A100 40GB device, while the update times for the RGB QR tasks reflect 8 A100 80GB SXM GPUs. When we used multiple GPUs, the update times were roughly 1/Num GPUs of the original time with a single GPU.

Table 2: Table shows various measures of the parallel computation demands for each environment.

| TASK | $|\pi|$ | UPDATE TIME (S) | GPU NOTES |
|---|---|---|---|
| FOUR ROOMS NAV | 386 | 37 | 1 A100 40GB OR 2 H100 80GB SXM5 |
| FOUR ROOMS PICK | 484 | 56 | 1 A100 40GB OR 2 H100 80GB SXM5 |
| RGB QR NAV | 2528 | 10 | 8 A100 80GB SXM |
| RGB QR PICK | 2528 | 10 | 8 A100 80GB SXM |

# D  ENVIRONMENT DETAILS

1. Stochastic Four Rooms Navigation
   - State dim: 2
   - Action space: Continuous
   - Action Dim: 2
   - Action range per dimension: $[-1, 1]$ reflecting $(\Delta x, \Delta y)$ for position of agent
   - $p(s_0)$ is a single $(x, y)$ position
   - $n = 5$ primitive actions

2. Stochastic Four Rooms Pick-and-Place
   - State dim: 4
   - Action space: Continuous
   - Action Dim: 4
   - Action range per dimension: $[-1, 1]$. First two dimensions reflect $(\Delta x, \Delta y)$ change in position for agent and the second two dimensions reflect the change in position for the object. The object can only be moved by the amount specified in the final two dimensions of the action if the object is within two units.
   - $p(s_0)$ is a single $(x_{\text{agent}}, y_{\text{agent}}, x_{\text{object}}, y_{\text{object}})$ start state
   - $n = 5$ primitive actions

3. RGB QR Code Navigation
   - State dim: 2
   - Action Dim: 2
   - Action space: Discrete
   - Action Range: $[-1, 1]$. First dimension reflects the horizontal movement. If first dimension is in range $\in [-1, -\frac{1}{3}]$, agent moves left. If first dimension is in range $[\frac{1}{3}, 1]$, agent moves right. Otherwise the agent does not make a horizontal movement. The second dimension reflects the north-south movement following the same pattern.
   - The RGB color vector for the colored squares in the QR code background is a 3-dim vector, in which each component is randomly sampled from the range $[0.7, 1]$ (i.e., has a light color). The agent is shown with a 2x2 set of black squares.
   - $p(s_0)$ is a single start state in the center of the room with a white background
   - $n = 5$ primitive actions

4. RGB QR Code Pick-and-Place
   - State dim: 4
   - Action Dim: 4
   - Action space: Discrete
   - Action Range: $[-1, 1]$. First two dimensions are same as navigation task. The second two reflect how the object will be moved provided the object is within two units.
   - The RGB color vector for the colored squares in the QR code background is a 3-dim vector, in which each component is randomly sampled from the range $[0.7, 1]$ (i.e., has a light color). The agent is shown with a 2x2 set of black squares. The object is shown with a 2x2 set of yellow squares.

- $p(s_0)$ is a single start state in which the agent and object are in same position in the center of the room with a white background
- $n = 5$ primitive actions

5. Continuous Mountain Car
   - State dim: 2
   - Action space: Continuous
   - Action Dim: 1
   - Action range per dimension: $[-1, 1]$
   - $p(s_0)$ is a single x position and velocity.
   - $n = 10$ primitive actions

# E    IMPLEMENTING HIERARCHICAL AGENTS WITH SKILLSET EMPOWERMENT

Coding hierarchical agents that use the $(\phi, \pi)$ Skillset Empowerment skillsets as a temporally extended action space is simple. For the higher level policy $\pi : \mathcal{S} \to \mathcal{Z}$ that outputs a skill $z$ from the learned $(\phi, \pi)$ skillset given some state, attach a tanh activation function to this policy, which bounds the output to $[-1, 1]$, and then multiply that output by $\phi$, which will bound the skill action space to $[-\phi, \phi]$ in every dimension, which has the same shape as $d$-dimensional cubic distribution that $\phi$ represents. (Note that in our implementation, $\phi$ is technically the log of the half length of each side of the $d$-dimensional cubic uniform distribution so the output of the tanh activation function should be multiplied by $e^\phi$. We have $\phi$ represent the log of the half length of each side so the $\phi$ actor $f_\mu$ can output negative numbers.) Then once a skill $z$ has been sampled, the skill can be passed to the Skillset Empowerment skill-conditioned policy $\pi(a|s_0, z)$ which will then execute a closed loop policy in the environment.

# F    BASELINE DETAILS

For the GCRL comparison, in the low-dimensional stochastic four rooms domains, we compared against the variant of GCRL that is a lower bound to Empowerment, in which the variational posterior consists of a tight diagonal gaussian centered at the skill-terminating state. The goal distribution is set to the distribution of all reachable state (e.g., all possible agent $(x, y)$ positions in the stochastic four rooms navigation task). For the higher dimensional QR code tasks, we implemented Reinforcement learning with Imagined Goals (RIG) Nair et al. (2018). RIG is an unsupervised GCRL method that combines representation learning and GCRL. RIG uses a VAE to separately learn an encoder that maps states to distributions over skills and a decoder that maps latent states to distributions over observations. RIG then performs GCRL in the learned embedding space (i.e, the agent learns skills that target specific latent states). Because the focus of this paper is not exploration, we make it easier on the representation learning component of RIG and provide it with a large dataset of reachable observations (e.g., images of the agent and object in a large variety of positions in the pick-and-place QR code environment.) The goal distribution for the GCRL phase is the prior distribution $p(z)$ from the VAE component.

## G   MUTUAL INFORMATION ENTROPY VISUALIZATIONS

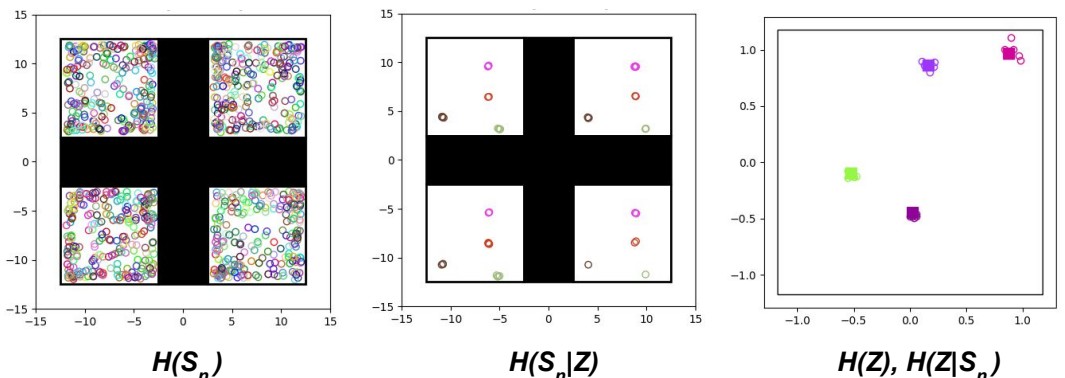

Figure 6: Entropy visualizations for the stochastic four rooms navigation setting. Left image visualizes $H(S_n)$ by marking the skill-terminating state from 1000 skills randomly sampled. The center image visualizes $H(S_n|Z)$ by showing 12 samples of skill-terminating states from 4 specific skills randomly sampled. The right image visualizes (i) $H(Z)$ by showing the square skill distribution $\phi$ (black square outline) and (ii) $H(Z|S_n)$ by showing samples of the variational posterior (empty circles) for four different skills (filled squares)). The images show the agent has learned a large abstract skillset, in which different skills target different $(x, y)$ offset positions.

**Stochastic Four Rooms Navigation – Pre-Training**

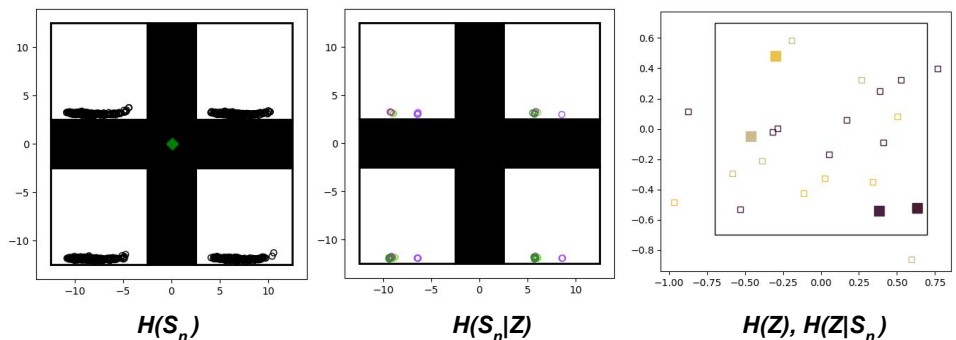

Figure 7: Entropy visualizations for a **non-trained** Skillset Empowerment agent in the stochastic four rooms navigation task. Per the poor state coverage in the left image and the high entropy posterior distributions in the right image, the agent does not start with a diverse skillset.

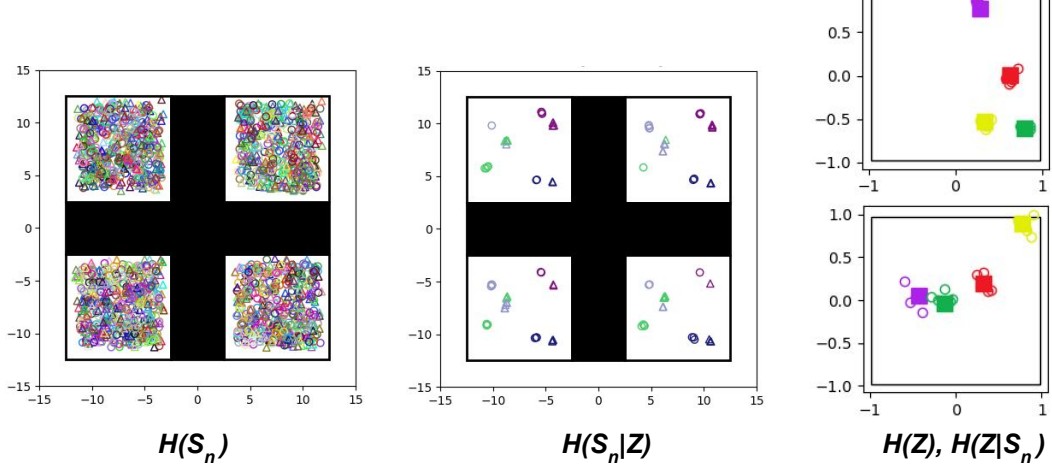

Figure 8: Images show the entropy visualizations for the stochastic four rooms pick-and-place domain. The near uniform coverage of the state space in the left image shows that $H(S_n)$ is large. Note that the circles represent the final $(x, y)$ position of the agent, and the triangles represent the final $(x, y)$ position of the object. The middle image visualizes $H(S_n|Z)$ by focusing on four skills, uniformly sampled from the skill space, and for each skill displaying 12 samples of skill-terminating states $s_n$. Per the image, each skill targets an abstract state representing an offset from the center of a room for both the agent and object. For instance, the blue skill results in the agent carrying the object to the bottom right corner of the room. The right image focuses on four skills and shows 5 samples from the variational posterior $q_\psi(z|s_0, l, \theta, s_n)$. Per the image, the samples form a narrow distribution around the executed skill, showing that $H(Z|S)$ is low.

**Stochastic Four Rooms Pick-and-Place – Pre-Training**

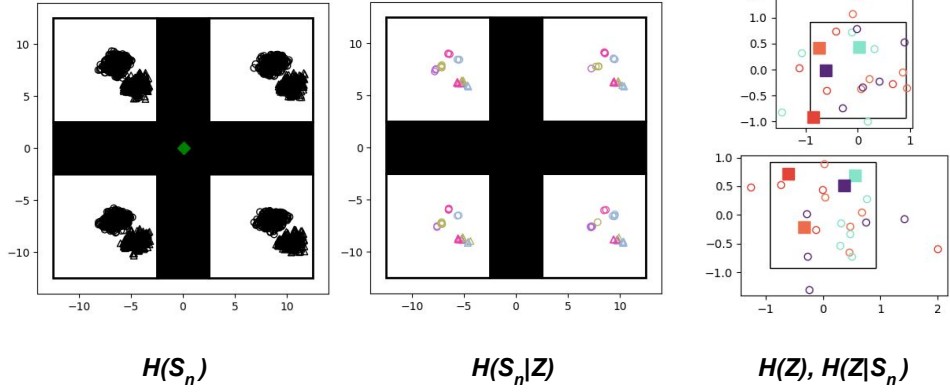

Figure 9: Entropy visualizations for a **non-trained** Skillset Empowerment agent in the stochastic four rooms pick-and-place task. Again, per the poor state coverage in the $H(S_n)$ visual and the high entropy posteriors in the $H(Z|S_n)$ visual, the agent does not start with a diverse skillset.

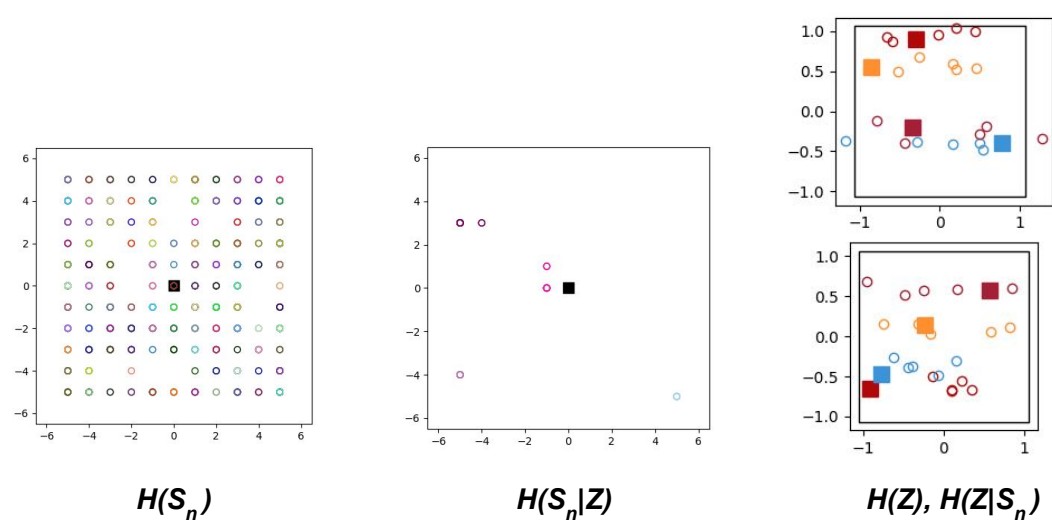

$H(S_n)$ $\quad\quad\quad$ $H(S_n|Z)$ $\quad\quad\quad$ $H(Z), H(Z|S_n)$

Figure 10: Entropy visualizations for the RGB QR code navigation task. Left image visualizes $H(S_n)$ by marking the skill-terminating states $s_n$ produced by executing 1000 samples of skills from the learned skill space. Center image visualizes $H(S_n|Z)$ by executing four skills 12 times each and recording the skill-terminating states. Each skill targets an abstract (x,y) position. Note the agent does not have access to the underlying state (i.e., the $(x, y)$ position of the agent) that is marked. The agent receives a 432-dim state (i.e., a 12x12x3 image). The right image shows samples from the variational posterior distribution. Note that in this case, the latent space is four dimensional even though the underlying state space is two dimensional. Because the agent does not need those extra dimensions, you see the horizontal lines in the variational posterior visualization.

**RGB QR Code Navigation – Pre-Training**

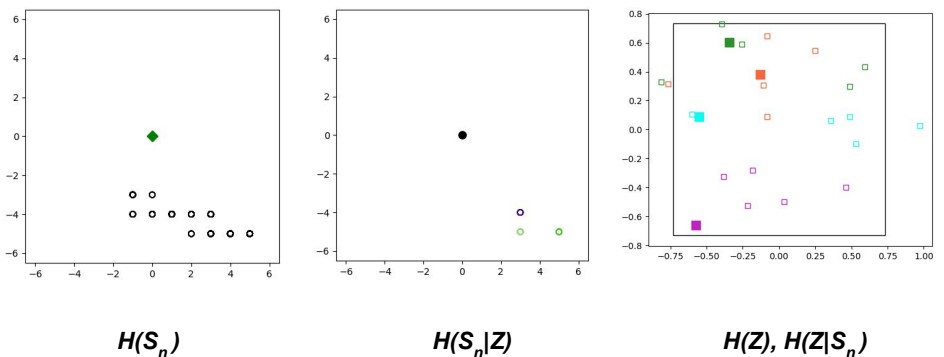

$H(S_n)$ $\quad\quad\quad$ $H(S_n|Z)$ $\quad\quad\quad$ $H(Z), H(Z|S_n)$

Figure 11: Entropy visualizations for a **non-trained** Skillset Empowerment agent in the RGB QR code navigation task. Per the visuals, the agent does not start with a diverse $(\phi, \pi)$ skillset.

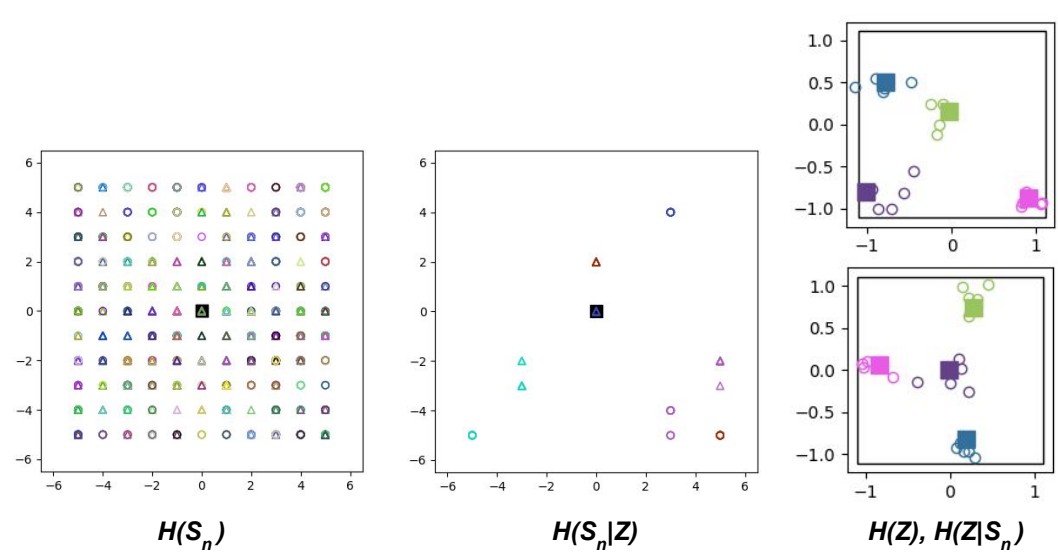

Figure 12: Entropy visualizations for the RGB QR code pick-and-place tasks. Left image visualizes $H(S_n)$ by marking the skill-terminating states $s_n$ produced by executing 1000 samples of skills from the learned skill space. Note that the circles represent the final $(x, y)$ position of the agent, and the triangles represent the final $(x, y)$ position of the object. Center image visualizes $H(S_n|Z)$ by executing four skills 12 times each and recording the skill-terminating states. Each skill targets an abstract (x,y) position for both the agent and object. The right image shows samples from the variational posterior distribution. Per the visuals, as expected, $H(S_n)$ is large while the conditional entropies $H(S_n|Z)$ and $H(Z|S_n)$ are small.

**RGB QR Code Pick-and-Place – Pre-Training**

Figure 13: Entropy visualizations for a **non-trained** Skillset Empowerment agent in the RGB QR code pick-and-place task. Per the visuals, the agent does not start with a diverse $(\phi, \pi)$ skillset.

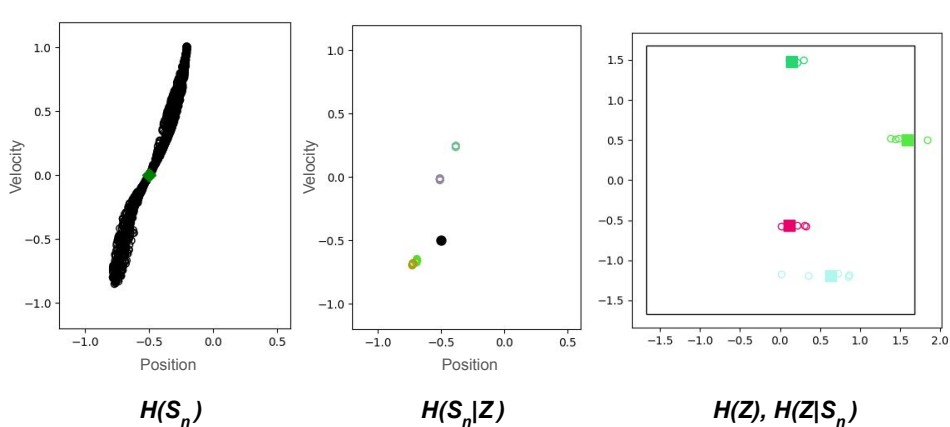

Figure 14: Entropy visualizations for a Skillset Empowerment agent agent in the Continuous Mountain Car task. The x-axis in the $H(S_n)$ and $H(S_n|Z)$ visuals show the agent position component of $s_n$ and the y-axis shows the velocity component of $s_n$. The black dot in the $H(S_n|Z)$ shows the starting state for the mountain car agent. Per the images, the agent has learned a diverse skillset, in which skills target different tuples of (cart position, cart velocity).

**Continuous Mountain Car – Pre-Training**

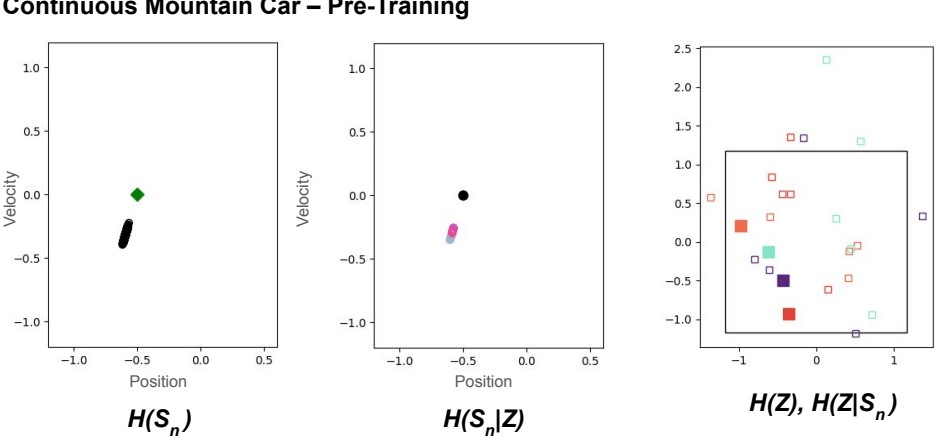

Figure 15: Entropy visualizations for a **non-trained** Skillset Empowerment agent in the Continuous Mountain Car task task. Per the visuals, the agent does not start with a diverse $(\phi, \pi)$ skillset.

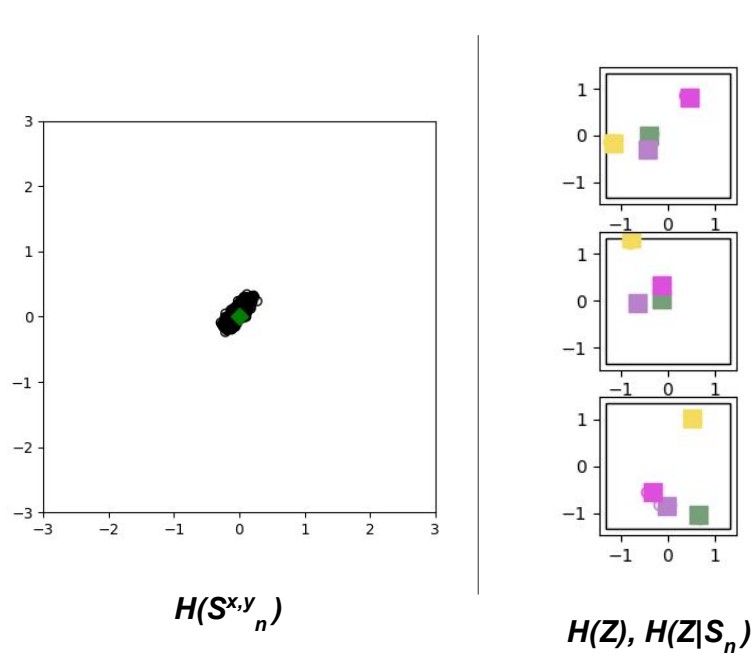

Figure 16: Applying Skillset Empowerment to the large ant domain produced some interesting results. The agent learned a massive skillset of 16.8 nats, which is around 20 million skills. The $H(Z), H(Z|S_n)$ visual shows these are distinct skills as the variational posterior distributions $q_\psi(z|s_0, \phi, \pi, s_n)$ are tight. On the other hand, these skills do not involve much torso translation as shown by the $H(S_n^{x,y})$ visual which plots the $(x, y)$ torso positions from $s_n$.

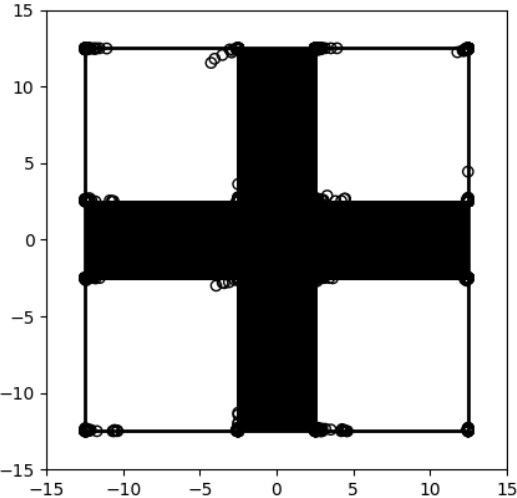

Figure 17: GCRL state coverage in stochastic four rooms domain. The agent only learns skills to target the corners of the rooms.

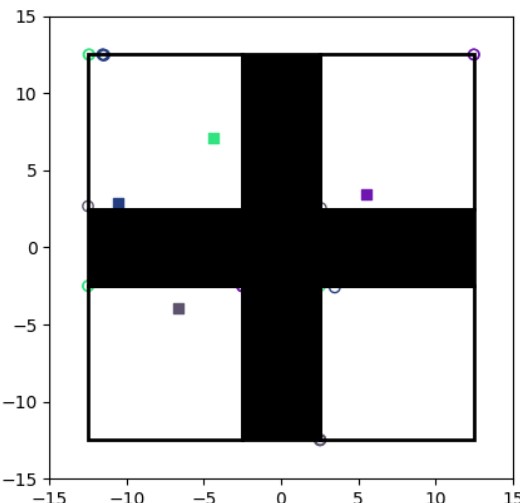

Figure 18: Image shows the skill-terminating states (empty circles) from four randomly selected goal states (filled squares). Each skill simply moves in the direction of the room of the goal state regardless of where in the room the goal is. For instance, given the purple goal (shown by purple square) in the lower left of the top right room, the agent just moves to the top right, which you can see by the purple circles in the top right room and bottom left room (hard to see). Similarly for the dark blue goal, the agent just moves to the top left, which you can see by the blue circles in the top left of the various rooms.

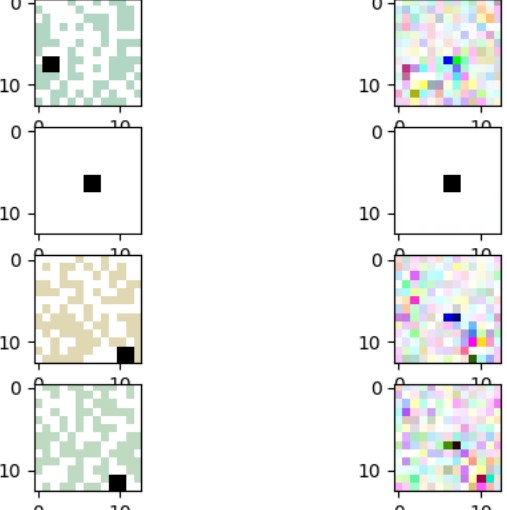

Figure 19: Image shows a sample of the VAE results in the RGB QR code navigation task. The left column shows sample images from the environment and the right column shows the results when those samples are encoded and then decoded. The VAE was able to decode the initial state of the environment, which is just a white background with the agent in the center, but struggled for other states.

