# OpenReview forum: "Learning Large Skillsets in Stochastic Settings with Empowerment"
_ICLR.cc/2025/Conference — Submitted to ICLR 2025_

### Official Review · Reviewer_HA6E · 2024-10-28

**Soundness:** 2
**Presentation:** 1
**Contribution:** 1
**Rating:** 3
**Confidence:** 5

**Summary:**

This paper studies how to improve the approximation of the empowerment objective for unsupervised skill learning in the domain of Unsupervised Reinforcement Learning (URL).  It highlights limitations in previous approaches due to loose lower-bound approximations and proposes a method to tighten these bounds. The proposed method is primarily evaluated in low-dimensional environments.

**Strengths:**

1. This paper offers an interesting perspective on revisiting the approximation of the classic unsupervised skill learning objective $I(S;Z)$,
2. The proposed approach is generalizable and applicable to most existing URL methods that approximate the conditional state distribution $q_\psi(S|Z)$.

**Weaknesses:**

> ## Regarding motivation and contribution

In Section 2.2

*"Several prior works have empirically demonstrated this result in which existing empowerment approaches tend to learn skillsets that do not change much from initialization."*

The previous empirical findings can not justify that the issues are caused by the loose lower bound approximation (this paper's focus). Instead, [1][2] consider the issues are implicit for $I(S;Z)$ objective because it can be optimized by skillsets close to initialization.

**This paper contributes only to the approximation level of the $I(S;Z)$ objective**, so it also suffers the same implicit issue of $I(S;Z)$ as shown in Figure 16 (the torso positions of Ant remain close to initialization)

Furthermore, the focused problem can be considered as $\max_{\psi,\pi} \tilde{I}(\psi,\pi)$, where $\psi$ being the parameter for the variational posterior $q_\psi(Z|S)$. The difference between prior work and this paper is that existing methods update $\pi$ and $\psi$ alternatively in different time steps, while this paper updates $(\pi,\psi)$ together in one single time step. Figure 1 (right) only shows the single-step update of $\pi$ thus could be misleading to leave an impression that existing methods can never learn skillsets such as $\pi_1$. In fact, existing methods that alternatively update $\pi$ and $\psi$ could still learn skillsets similar to $\pi_1$.

In URL research, the notion of diversity is highly related to downstream task adaptation. Skills do not need to cover the entire state space to be diverse, as demonstrated in [8]. A smaller number of skills can still achieve diversity if they occupy distinct regions within the state space, with each skill serving as an effective fine-tuning initialization for downstream tasks within its specific region. Therefore, skill count alone is not an effective measure of skill diversity. Given the significant computational resources involved, the resulting improvements in downstream task adaptation may be minimal.

The overall motivation and contribution are more like findings instead of pushing the boundaries of URL research, making it more appropriate for a workshop.

> ## Regarding methodology

The need to search across the model parameter space makes this approach computationally intensive, even for simple 2D implementations. Using LLM-level computational resources for relatively impractical URL problems is extremely inefficient, especially given that the core contribution of this paper is a shift from alternating updates of $\pi$ and $q_\psi$ to simultaneous updates of $(\pi,q_\psi)$. Additionally, the paper provides no evidence that state of the art URL methods could not achieve similar skill diversity with significantly lower computational demands.

> ## Regarding evaluation

Experiments are mainly performed in simple low-dimensional environments, baselines are old and outdated. The number of skills is not compared to more recent methods such as [1][2][3][4]. It is also unclear how the number of distinct skills is calculated.
There are no $H(Z), H(Z|S_n)$ figures for VIC to compare the core contribution of tightening the KL divergence bound between the true and variational posterior.

> ## Regarding rigor

The paper frequently discusses "diversity" without a rigorous definition. It seems that in this paper more diversity means more distinct terminating states. If so, $I(S;Z)$ is not a good measure of diversity. Maximizing $I(S;Z)$ can only guarantee that the distribution of $p(S|z)$ has large KL divergence to average state distribution $p(S)$ ([5]), but can not guarantee a large KL-divergence $D_{KL}(p(S|z_1)||p(S|z_2))$ between the conditional state distributions of different skills ([6]).

As mentioned before, a limited number of skills can also exhibit diversity and achieve comparable downstream task performance with significantly fewer pretraining resources. This requires the terminating states to be not only distinct but also sufficiently distant from one another, Wasserstein distance-based metrics ([6][7]) are better choices to measure diversity in this sense.

It is not clear how the number of distinct skills is evaluated in the experiments. While the metric $\tilde{I}(Z;S_n)$ was mentioned, the exact connection between this mutual information measure and the actual number of skills remains unclear.

> ## Regarding presentation

Section 1 has overly wordy explanations of the math formula, which makes the reading uncomfortable. Since the relevant formulas appear in Section 2 (e.g., (5)-(7)), it would be clearer and more efficient to introduce these formulas earlier.

Overall, the text is overly wordy without delivering a proportional amount of information. The reading requires considerable focus and energy, yet feels underwhelming in terms of insights gained.

> ## References





*[1] Lipschitz-constrained unsupervised skill discovery.*

*[2] Controllability-aware unsupervised skill discovery*

*[3] APS: Active Pretraining with Successor Features*

*[4] Contrastive Intrinsic Control for Unsupervised Skill Discovery*

*[5] The Information Geometry of Unsupervised Reinforcement Learning*

*[6] Task Adaptation from Skills: Information Geometry, Disentanglement, and New Objectives for Unsupervised Reinforcement Learning*

*[7] Wasserstein Unsupervised Reinforcement Learning*

*[8] Explore, Discover and Learn: Unsupervised Discovery of State-Covering Skills*

**Questions:**

1. See weaknesses

2. How exactly is the number of distinct skills evaluated? Are $q_\psi(z|S)=1$ for every skill $z$ after traning?

3. How long is the total training time with the computation demands in Appendix B?

4. How sensitive is the training process to hyperparameters?



***
## After detailed discussions
After detailed discussions with the author, I found that I had no misunderstanding from the conceptions to the theories, and I gained a better understanding of the limitations of the implementations.  I can confirm that my rating of 3 and confidence of 5 is reasonable.

**Details Of Ethics Concerns:**

There are no explicit indications of the author's identity. However, Section 3.3 includes information about the author’s other publications, which could indirectly reveal their identity. Though not immediately obvious, this information could still pose a risk to anonymity.

---

> ### Comment · Reviewer_HA6E · 2024-11-21
>
> Since there has been no rebuttal from the author, and another reviewer gave a rating of 8, I am considering lowering my rating to convey that, despite the significant amount of work put into the paper, the ultimate outcome appears to be a considerable waste of resources for no significant contribution.  This should not be encouraged.

---

> ### Author Response · Authors · 2024-11-21
> **Rebuttal**
>
> Page 1 of 2
>
> Thank you for your feedback and apologies for the delay.
>
> - Contribution is limited as only changing search procedure for (skill-conditioned policy $\pi$, variational parameters $\psi$)
>
> We do not believe that characterizing our contributions as limited is accurate.  Our first contribution of proving that existing approaches (e.g., DIAYN) only maximize loose lower bounds on the mutual information between skills and states should be important to the community as it provides a new and principled reason for why other papers (e.g., (i) Campos et al., 2020 and (ii) Levy et al., 2023) have shown that existing approaches typically only learn small skillsets.  (Note that the simple proof for why our approach can learn a tighter bound on the empowerment of a state is provided in section A of the appendix in the updated draft.  The proof was previously left out because we felt the wording we provided in the original manuscript was sufficient).  In addition to this contribution, we then provide an alternative objective that provides a tighter lower bound to the mutual information between skills and states.  Yes, this objective just involves training a variational posterior conditioned on the skill-conditioned policies and distribution over skills that are being considered, but it provides a more principled way to maximize mutual information.  Further, we also introduce a new actor-critic architecture to optimize this tighter variational lower bound as it is not clear how to maximize this new objective with existing techniques.  Empirically, we also provide the first results in a stochastic setting for an algorithm that maximizes the mutual information between skills and states.  We believe these contributions should be sufficient for a conference paper.
>
> - $I(Z;S)$ is “not a good measure of skillset diversity”, “unclear relationship to number of skills”,
>
> These comments are not true.  The mutual information between skills and states given (i) a particular distribution of skills $\phi$ and (ii) a skill-conditioned policy $\pi$ computes the number of skills from the ($\phi,\pi$) skillset that can be executed and decoded from the skill-terminating state with arbitrarily low error (Cover & Thomas, 2006).  Note, that this number of skills is a rate (i.e., the number of skills per use of the channel between skills and states).  Thus, maximizing mutual information will produce skills that target different skill-terminating states (i.e., will optimize skillset diversity) because the skills need to be decoded with arbitrarily low error.  Mutual information can also thus represents the “number of skills” in a skillset.  When we define “number of skills” as mutual information, we can emphasize that this is a rate (number of skills per channel use).  Also, we are not the first to use variational mutual information between skills and states to mean “number of skills” (e.g., Gregor et al. 2016).
>
> - Results show $I(Z;S)$ has an “implicit issue”
>
> I am not sure what “implicit” issue you are referring to and none of our results produce a skill-conditioned policy that was similar to the initial skill-conditioned policy.  In the one example chosen in the review, Ant, the agent went from having a minimal skillset to having a massive skillset with ~19.7 million skills (i.e., a mutual information of 16.8 nats).  Although these skills did not produce much translation they produced specific joint movements.
>
> - Computationally expensive approach requiring “LLM-equivalent” compute for simple tasks
>
> A few comments here.  First, the reviewer takes issue with the approach of searching over different distributions of skills and skill-conditioned policies.  But mutual information depends on these two distributions, so searching over these two distributions to maximize mutual information is principled.
>
> Second, yes the approach requires extensive compute, but we are learning large skillsets with this compute.  Sure you can argue that many of these skills could be unnecessary.  But there also may be many situations in which a lot of these skills are necessary.  For instance, there may be a setting where an ant agent needs to be able to open a variety of door handles.  In this case, the ant agent may need to achieve states in which one of its legs is at a certain angle moving with a certain angular velocity.  In this type of situation, the agent would need to have skills that are not limited to (x,y) translation.  In addition, for downstream applications such as empowerment-based reinforcement learning where an agent takes actions to maximize an empowerment reward, the empowerment of states needs to be known so agents need to know all the skills they have in a state.  Unsupervised skill-learning Algorithms like LSD and CSD that do not actually maximize mutual information would not be sufficient for this application.

---

> > ### Author Response · Authors · 2024-11-21
> > **Rebuttal (Page 2 of 2)**
> >
> > Page 2 of 2
> >
> > Third, characterizing the level compute we used for our simpler tasks as “LLM-equivalent” is not true.  Our easiest tasks required 4 H100s for 1-5 hours, which from my understanding is not near the amount of compute required to train LLMs.  This criticism also implicitly assumes the price of compute will not fall over time, which has not been true historically.
> >
> > - Missing comparisons to LSD, CSD, CIC
> >
> > As mentioned before, comparing to LSD and CSD would not be an apples to apples comparison.  These approaches learn smaller skillsets than our approach as they do not maximize the mutual information between skills and states, and thus can be missing skills/cannot be used for downstream applications like empowerment-based reinforcement learning in which an agent needs to maximize an empowerment reward.
> >
> > CIC uses another way to approximate the lower bound on the mutual information between skills and states instead of a maximum likelihood objective like DIAYN and our approach.  But because CIC does not approximate this lower bound for other skill-conditioned policies that are being considered (i.e., the reward does not change for different policies), CIC is still maximizing a loose lower bound on the mutual information between skills and states.  CIC also does not provide the actual learned mutual information learned from what we can see, so it’s unclear how well it is truly maximizing mutual information between skills and states.
> >
> > - How exactly is number of skills evaluated?
> >
> > We use the variational mutual information between skills and states for this number.  Again, this is principled because the mutual information between skills and states represents the number of skills (per use of the channel) that can be executed and decoded with arbitrarily low error (Cover & Thomas, 2006).
> > - How long is training time?
> >
> > For all tasks except Ant, training time was 1-5 hours. The navigation tasks would be on the low end of this range and the larger pick-and-place tasks would be near the upper end.  Ant, due to its much larger underlying state space, was trained for around three days.
> >
> > - How sensitive to hyperparameters is training?
> >
> > The algorithm does not have any out of the ordinary parameter dependencies.
> >
> > 1. Campos et al. Unsupervised Discovery of State-Covering Skills. 2020
> > 2. Levy et al. Hierarchical Empowerment: Towards Tractable Empowerment-based Skill Learning. 2023
> > 3. Gregor et al. Variational Intrinsic Control. 2016
> > 3. Cover and Thomas.  Elements of Information Theory. 2006

---

> ### Comment · Reviewer_HA6E · 2024-11-21
>
> Thanks for the reply, I want to make some comments on your reply:
>
> ## About contribution and "implicit issues ":
>
> 1. The bound still only contributes to the approximation level.  All the bounds in Appendix A come from that (8) is a better approximation than (7).
>
> 2. Directly comparing (8) with (7) is unfair and potentially misleading. As I mentioned in my original review, existing methods that use (7) alternatively update $(\pi,\phi)$ and $\psi$ and it is totally capable of learning $\pi_1$ in Figure 1. There are two steps for each update, first update $(\pi,\phi)$ with (7) then update $\psi$ with (6).  Directly comparing (8) with (7) is like comparing $I(\pi_{k+1},\phi_{k+1},\psi_{k+1})$ with $I(\pi_{k+1},\phi_{k+1},\psi_k)$, insufficient to show the bound's long-term significance as $k\rightarrow \infty$. In optimization literature, block coordinate descent[3](alternative update) is possible to converge to the same solution as full gradient descent(simultaneous update).
>
> 3. Last but not least, you seem to not fully understand the most important point, the “implicit issue” of $I(S;Z)$. I was clearly referring to [1][2]. You should read them before you make further comments. Even if you spend more effort to make a perfect approximation of $I(S; Z)$, fundamental limitations persist. This undermines the significance of your contribution, despite your substantial effort.
>
> ## About diversity:
> This concern is not addressed. Your response just states a well-known property of mutual information, but you are unaware of the more recent theoretical results in Unsupervised RL and what "diversity" means for good downstream task adaptation. I can still use my initial review about "rigor" to reply to your rebuttal.
>
> ## About skill number:
>
> (Cover & Thomas, 2006) is a textbook and I am not sure what you mean by referring to it.
>
> There is a reason why I asked are you assuming $q_\psi(z|S)=1$ after training. Because only when $q_\psi(z|S)=1$, $\hat{I}(S|Z)=E[\log \frac{q_\psi(z|S)}{\phi(Z)} ]=E[-\log \phi(Z)]$ is log of skill number. However, I did not see how this assumption could be justified.
>
> ## About computation:
> Then what is the training time for existing methods in the same setting? If you are training in parallel, then what is the memory usage?
>
> ## About baselines:
> The formulation of LSD and CSD does not directly resemble $I(S; Z)$ because they are aware of the limitations of $I(S; Z)$ and proposed new objectives to overcome them.
> CIC also saw the limitation and they optimized the $I(\tau; Z)$, where $\tau$ is the trajectory. Despite the approximation bound, it could result in much more diverse behavior because the space of trajectories is inherently much larger than the state space.
>
> The CIC paper is not accepted because there might be potential inconsistencies between their reported results and their code. However, LSD and CSD can still serve as strong baselines, particularly for higher-dimensional control environments like Ant.
> ***
> I strongly recommend that you stay updated with the latest research in this area. I apologize if my language seemed overly direct, and I truly appreciate the effort you've put into this work. However, I believe it might be beneficial to reconsider the current research direction in light of these recent developments.
>
>
> ***
> [1] Lipschitz-constrained unsupervised skill discovery.
>
> [2] Controllability-aware unsupervised skill discovery
>
> [3] Wright S J. Coordinate descent algorithms

---

> > ### Author Response · Authors · 2024-11-22
> >
> > Thank you for your comment.  I believe both of your two main points are incorrect.  First, I reject the position that the mutual information objective $I(Z;S|\pi)$ (I'll ignore $\phi$ to simplify) can be replaced with the the variational mutual objective $f(\pi,\psi)$, in which variational posterior parameterized by $\psi$ replaces the true posterior.  This would mean that $I(Z;S|\pi) = f(\pi,\psi)$ for all $\pi$, which is clearly incorrect.  They are only equal if $D_{KL}(\text{true posterior}||\text{variational posterior})=0$.  Skillset Empowerment in theory ensures this KL=0 so $f(\pi,\psi)$ can replace $I(Z;S|\pi)$.  I also do not understand your position that existing approaches would perform well in the example in Figure 1.  With existing approaches, skillset candidate 1 would have an undefined variational mutual undefined or variational mutual information of $-\infty$ because $p(z|s_n,\pi_{\text{Current}})$ either does not exist (if $\pi_{\text{Current}}$ does not target $s_n$) or is 0.
> >
> > Second, I have read the works of Park et al. and understand his criticism that $I(Z;S)$ is not useful because it can be maximized by having each skill target a infinitesimal point of some small region of the state space.  This is true in completely deterministic environments (deterministic policy and dynamics) when $\log$ posterior can go to infinity.  However, in practice this problem can be easily overcome by adding a small amount of stochasticity to the policy (e.g., in our implementation we used a diagonal gaussian policy with a fixed small standard deviation).  After this change, in practice you will not get infinite posteriors because there will be some overlap in the states targeted by different skills.  Our results in continuous mountain car and results we do not show like in simple 2D point domains show this is not a problem.  Agents will learn skillsets that cover the state space and are not limited to some small region.

---

> > > ### Comment · Reviewer_HA6E · 2024-11-22
> > >
> > > What I was saying from the beginning is that with $k\rightarrow \infty$, existing methods are capable of learning $\pi_1$. However, you are still looking at the effect of one single step.
> > >
> > > About the example, $f$ could be $\tilde{I}_{existing}$，
> > >
> > > As  $k\rightarrow \infty$, variational posterior gets closer to true posterior, and $\tilde{I}_{existing}$ gets closer to $I$.
> > >
> > > My point is that as $k\rightarrow \infty$, the benefit of learning a much more expensive SE objective is unknown, because no valid comparison against SOTA methods is provided.
> > >
> > > Your second point did not convince me. If so, why don't you compare them as baselines for mountain car and Ant?
> > >
> > > ***
> > > The responses from the author have been deflecting focus from the core problems:
> > > * Does skill numbers matter? How about the quality of the skills? Is the measurement for skill number even rigorous?
> > > * Is it worth the computational resources just to get a better approximation of an outdated objective?
> > > ***
> > > Therefore, further discussion is unnecessary. Moreover, I don't have unlimited time and I have been more responsible than most reviewers. I will keep my current rating and wait for the discussion to end.

---

> > > > ### Author Response · Authors · 2024-11-22
> > > >
> > > > Thank you for the response and I appreciate the back and forth discussion.  I address each of your criticisms below, and I understand if you do not want to discuss further.
> > > >
> > > > Regarding the Skillset Empowerment objective and the existing mutual information objective (e.g., DIAYN), two things can be true at the same time.  (i) Both objectives can have the same optimal pair of skill-conditioned policy and variational parameters $(\pi^*,\psi^*)$, while (ii) the existing mutual information objective between skills and states can suffer from local maxima with respect to the skill-conditioned policy.  If existing mutual information objective did not suffer from poor local maxima, VIC/DIAYN would be sufficient and we would not need the several algorithms that followed including the works of Park et al. and our work, Skillset Empowerment.  But we do not have empirical evidence that DIAYN can consistently learn large skillsets and have a lot of empirical evidence that the opposite is true (see Gregor et al. 2016 (explicit VIC nearly same as DIAYN), Campos et al. 2020,  Levy et al., 2023, and our work here).  Further, as our theory shows, maximizing the DIAYN objective with respect to $\pi$ is equivalent to maximizing a loose lower bound on the mutual information between skills and states for $\pi$ that differ from $\pi_{\text{Current}}$, which can encourage the agent to not change its policy (i.e., there can be a local maximum with respect to $\pi$).  On the other hand, in our work we demonstrate that Skillset Empowerment does not suffer from these local maxima.
> > > >
> > > > I still do not understand the criticism that the channel capacity, or the maximum mutual information between skills and states, is not a rigorous way to define the number of distinct skills in a skillset $\pi$.  The number of distinct skills that can be executed per use of the channel is exactly what the channel capacity of this channel is.
> > > >
> > > > The reviewer also makes the criticism that the number of skills that an agent can execute from a state is not important.  As I mentioned before, for the application of empowerment-based reinforcement learning (Klyubin et al., 2005), in which agents maximize an empowerment reward (i.e., take actions to go to states where the agent has more skills), the maximum mutual information is needed and the works of Park et al. would not be sufficient.  I also don't see how there can be a guarantee that for the vast set of tasks we would want a general purpose agent to do, agents would not need skills that can target each of (or at least a vast majority of) the reachable states.
> > > >
> > > > The last criticism of the reviewer is that computing mutual information is expensive.  Yes this is true and should not be surprising.   The mutual information between skills and states of a skill-conditioned policy computes the number of distinct skills contained in the skill-conditioned policy, which requires executing those skills in the environment to see the states they target.  What seems surprising to me are the claims of existing work that the number of distinct skills in a skill-conditioned policy can be measured without the expensive operation of executing a large number of skills in the environment.
> > > >
> > > > Gregor et al.  Variational Intrinsic Control. 2016
> > > >
> > > > Campos et al. Explore, Discover, Learn. 2020
> > > >
> > > > Levy et al. Hierarchical Empowerment: Towards Tractable Empowerment-based Skill Learning. 2023
> > > >
> > > > Klyubin et al. All Else being Equal be Empowered. 2005

---

> > > > > ### Comment · Reviewer_HA6E · 2024-11-23
> > > > >
> > > > > Your evaluation metric $\tilde{I}$ is even dependent on the parameterization of $q_\psi$, meaning that different parameterizations could result in different "skill numbers".
> > > > >
> > > > > What I have been criticizing was at the conceptual level, there are even more concerns at the implementation level:
> > > > >
> > > > > 1. The concern of reviewer mbwo about the expressiveness of your critics and variational posteriors is quite valid because it can not model the cooperative effect of changing multiple parameters together.
> > > > >
> > > > > 2. The concern of reviewer jSiL about requiring a dynamic model is valid.
> > > > >
> > > > > If you have an idea that overcomes the need for transition dynamics and is scalable, my suggestion is that you use it to learn more recent objectives like $\tilde{I}(Z;\tau)$. A convincing result to justify the additional computational demand is that you successfully demonstrate that you can find skills of both better quantity and quality than recent methods, and then efficiently find the skills needed for more complex downstream tasks.

---

> ### Author Response · Authors · 2024-11-24
>
> Thank you for your response.
>
> See below regarding your concerns about the implementation.
>
> 1. The gradient of the mutual information critic $Q(s_0,\pi)$, in which $\pi=f_{\lambda}(s_0)$ (ignoring $\phi$), with respect to $\lambda$ only requires the gradients of the mutual information critic with respect to each dimension of $\pi$.  That is, $\nabla_{\lambda} Q=\nabla_{\pi} Q \cdot \nabla_{\lambda} \pi = \sum_{i=1}^{|\pi|} \nabla_{\pi_i} Q \times \nabla_{\lambda} \pi_i$ (i.e., $\nabla_{\pi} Q \cdot \nabla_{\lambda} \pi$ is the dot product between (i) the gradients of the critic with respect to each parameter in $\pi$ and (ii) the gradients of each parameter in $\pi$ with respect to $\lambda$).  This means that the parameter-specific critics that only measure the changes in mutual information from changes to a single parameter are sufficient to know the gradient with respect to $\lambda$ and the critics do not need to approximate changes to multiple parameters.
>
> 2.  The need to either have or learn a model of the transition dynamics was why we attached our follow up paper that shows how mutual information can be maximized using (a) our actor-critic architecture and (b) only off-policy transition data $(s_t,a_t,s_{t+1})$.  This algorithm maximized a lower bound to a tight upper bound to the mutual information between skills and states, and this objective can be optimized with off-policy data.  The results from the off-policy algorithm matched the performance of Skillset Empowerment in all tasks despite not having or needing to learn a simulator of the environment.
>
> Despite the limitation of needing to learn a simulator, we still believe Skillset Empowerment is a sufficient contribution as it provides a different way to maximize the mutual information between skills and states.  Yes there are other ways to learn skillsets in an unsupervised methods (e.g., the works by Park et al. that the reviewer has mentioned), but these works do not maximize the mutual information between skills and states and there may be a need for agents to maximize this quantity and learn a large number of the available skills in a state (e.g., empowerment-based RL).  In addition, our approach is the first unsupervised skill learning algorithm to learn skills in stochastic settings, which does not seem minor.
>
> Further, Skillset Empowerment seems to provide a more principled approach to address the problem of nonstationarity that can hinder unsupervised skill learning approaches.  In unsupervised skill learning, the reward changes with changes to the skill-conditioned policy.  This does not affect Skillset Empowerment because the action is the skill-conditioned policy and so the action is a sufficient statistic for the reward.  But in other approaches this is not the case as the action is still the primitive action so each change to the policy will change the reward associated with a (state, skill, primitive action) tuple.
>
> 3. The variational mutual information performance metric we used should not be a problem.  The algorithms that we compared against, VIC and GCRL, should be learning policies in which the true posterior $p(z|s_n)$ is approximately diagonal gaussian so modeling the variational posterior as diagonal gaussian should not be problematic.  Sure it's possible there could be some noise in whether the gaussian variational posterior can match the true posterior, but this would affect all approaches and should disappear with more trials.  I also do not remember observing this in any trial.

---

> > ### Comment · Reviewer_HA6E · 2024-11-24
> >
> > Your point 1 is only true when you model $\nabla_{\pi_i}Q(s_0,\pi)$, you need to take parameters other than $i$ into the input.
> >
> > A simple example when $Q=\pi_1×\pi_2$, $\nabla_{\pi_1}Q=\pi_2$, but your formulation is not enough expressive for this.

---

> > > ### Author Response · Authors · 2024-11-24
> > >
> > > That's not the case if the critic $Q(s_0,\pi)$ is modeled as a neural network with layers in which a non-linear function is applied to a linear function of inputs.  In this case, the individual components of $\pi$ are weighted and summed and not multiplied together as in the example you provided.  The gradient of this $Q(s_0,\pi=f_{\lambda}(s_0))$ with respect to $\lambda$ would only depend on gradients of $Q$ with respect to individual $\pi_i$ components, which is why parameter-specific critics $Q(s_0,\pi_i)$ are sufficient for obtaining the same gradient as if the "full" critic $Q(s_0,\pi)$ were used.

---

> > > > ### Comment · Reviewer_HA6E · 2024-11-24
> > > >
> > > > This is not true even for a single-layer FNN with nonlinear activation function $a$ and two inputs, and one output.
> > > >
> > > > $$Q(\pi) = a(w_{1}\pi_1+w_{2}\pi_2)$$
> > > > $$\nabla_{\pi_1} Q(\pi) =\frac{ \partial a(w_{1}\pi_1+w_{2}\pi_2)}{ \partial (w_{1}\pi_1+w_{2}\pi_2)}w_1$$
> > > > Therefore, you still need the full $\pi$ as input for accurate $\nabla_{\pi_1} Q(\pi)$.
> > > >
> > > > Your claim only works for completely linear NN without activation functions, which inherently has huge limitations on expressibility

---

> > > > > ### Author Response · Authors · 2024-11-24
> > > > >
> > > > > The inner loop that trains $Q(s_0,\pi_i)$ assumes a skill-conditioned policy $\pi_i$ with (i) noisy values of the $i$-th dimension (i.e., $\pi_i[i] = f_{\lambda}(s_0) + \text{noise}$) and (ii) greedy values for the remaining dimensions (i.e., for $j\neq i, \ \pi_i[j] = f_{\lambda}(s_0)$) so the other dimensions of $\pi$ are not being ignored in $Q(s_0,\pi_i)$.  The point is that to determine the gradient of Q with respect to $\lambda$, only one dimension of $\pi$ needs to be changed at a time (i.e., we only need to measure $\nabla_{\pi_i}Q(s_0,\pi_i)$ for $i \in 1,\dots,|\pi|)$.

---

> > > > > > ### Comment · Reviewer_HA6E · 2024-11-24
> > > > > >
> > > > > > Your Eq.18 suggest you change every dimmension all together.
> > > > > > If you change it only one dimmension at a time, then it would suffer the exact local optimum issue that alternative updates/coordinate descend suffers.
> > > > > >
> > > > > > Anyway, this appears to be an inelegant fix for an overly computationally intensive algorithm. A better conceptual idea from the start would avoid resorting to such maximalism.
> > > > > >
> > > > > > This scalability limitation also reduces the potential of your method for future applications, especially those aiming to learn higher-quality skills.

---

> ### Author Response · Authors · 2024-11-24
>
> No all parameters of the skill-conditioned policy are being updated at the same time.  By "one dimension at a time", I was just referring to all the $Q(s_0,\pi_i)$ just approximating mutual information from changes to one parameter in $\pi$.
>
> The inner loop that trains $Q(s_0,\pi_i)$ is approximating a tighter lower bound to the mutual information $I(Z;S_n|\pi_i)$ so it will not have the same local max issue.  Note that the inner loop consists of a series of updates to first the variational posteriors $q_{\psi_i}(z|s_n,\pi_i)$ and then a series of updates to the $Q(s_0,\pi_i)$ functions.  Yes, this is more computationally expensive but should not have the same local max issues which our theory and empirical evidence demonstrate.

---

> > ### Comment · Reviewer_HA6E · 2024-11-24
> >
> > So you admit that your critic is not accurate for every $\pi$ and only provides accurate gradient for a current $\pi$.
> >
> > Then what's even the point of having so many neural network parameterized critics if they are only accurate around one $\pi$, why don't you just to nonparametric sensitivity analysis for your current $\pi$?

---

> > > ### Author Response · Authors · 2024-11-24
> > >
> > > No, each of the $|\pi|$ critics, $Q_i(s_0,\pi_i)$ for $i \in 1,\dots,|\pi|$, should accurately approximate $I(Z;S_n|\pi_i)$ for skill-conditioned policies $\pi_i$ that have small changes to the $i$-th parameter while the rest assume greedy values (i.e., the mutual information for way more than a single $\pi$ should be well-approximated).
> > >
> > > Yes I understand this is computationally expensive, but maximizing mutual information with respect to a skill-conditioned policy should be expensive because (a) to measure mutual information even for a single $\pi$, many skills $z \sim p(z)$ need to be executed and (b) the skill-conditioned policy has a large number of dimensions.

---

> ### Comment · Reviewer_HA6E · 2024-11-24
>
> It is only accurate around your current greedy policy,  nonparametric sensitivity analysis should be much more efficient for this.
>
> For example, a much more simplistic way of implementation is to sample a large number of $(\pi', \phi')$ around current $(\pi, \phi)$ and learn $\psi$ for all of them, evaluate their objective values and find the best one.
>
>
> ***
> My evaluation was done before the discussion on implementation details. The further discussion only serves as potential suggestions

---

> > ### Author Response · Authors · 2024-11-25
> >
> > It's accurate for the policies $\pi_i$ that it needs to find the gradient with respect to the skill-conditioned policy actor.  Re your suggestion for the nonparametric approach, either way a larger number of $(\phi,\pi)$ skillsets need to be tested and my guess is that losing the gradient isn't worth it on average, but sure this is a good idea to try as a baseline.
> >
> > Anyway, it seems like this discussion has reached the agree-to-disagree point.  The criticism that our approach requires more compute than existing approaches is true.  My guess though is that most existing unsupervised skill learning work will suffer from the local max issue because when the reward is only optimized for the current skill-conditioned policy there should be less of an incentive to change the skill-conditioned policy.   Our approach does not have this issue.  But we will see over time.

---

> ### Comment · Reviewer_HA6E · 2024-11-25
>
> Your implementation relies on the assumption that of certain smoothness of $Q(\pi)$. Without this assumption, you need to retrain all of your critic neural networks for every new greedy $\pi$, which is extremely inefficient. However, this assumption again limits the expressiveness of $Q(\pi)$.
>
> I suggest to look at techniques from zero-order(Derivative-free) optimization[1]. There should be efficient ways to sample and test $(\pi,\phi)$ skillsets.
>
>
> Also, many newer methods such as LSD,CSD and VISR(APS), are actually highly related to and much more intricate than what you mentioned:
> > " However, in practice this problem can be easily overcome by adding a small amount of stochasticity to the policy (e.g., in our implementation we used a diagonal gaussian policy with a fixed small standard deviation). After this change, in practice you will not get infinite posteriors because there will be some overlap in the states targeted by different skills."
>
> We can consider them as methods focusing on the "quality" of learned skills. My suggestion is that you find an approach with acceptable computational demand, learn skills with both better quantity and quality, and demonstrate its superiority against more recent methods. Achieving this would be a strong contribution to the URL community.
>
> ***
> [1] Larson et al. (2019): "Derivative-free optimization methods"

---

### Official Review · Reviewer_jSiL · 2024-11-01

**Soundness:** 3
**Presentation:** 1
**Contribution:** 3
**Rating:** 5
**Confidence:** 3

**Summary:**

This paper introduces Skillset Empowerment, a new approach aimed at enhancing skill diversity for agents operating in unpredictable environments. By tightening the mutual information bound between skills and states, this method allows agents to learn a wider range of distinct skills more effectively. Experimental results indicate that Skillset Empowerment outperforms prior approaches in complex, stochastic settings.

**Strengths:**

- Skillset Empowerment presents a fresh perspective on mutual information objectives, tightening the lower bound on mutual information between skills and states. This refined approach represents an original contribution to URL.
- The authors have conducted thorough experiments across multiple environments, which supports the empirical validity of their claims and provides a well-rounded assessment of the method's performance.

**Weaknesses:**

- Although the method successfully enables agents to learn a large skillset, the paper lacks evaluation on how these learned skills translate to practical, downstream tasks. The Unsupervised Reinforcement Learning Benchmark (URLB) [1] could be an appropriate choice to test this.
- The method assumes access to accurate transition dynamics, which restricts its applicability. This assumption is also problematic because, if the dynamics are known, planning and solving the MDP directly becomes feasible, reducing the need for URL methods.
- The computational demands are high, with individual critics trained for each parameter and multiple variational posteriors required. This increases the resource burden significantly compared to other URL methods, as shown in Appendix B.
- While the paper includes quantitative metrics, it falls short in qualitatively examining the distinctiveness and usefulness of the learned skills. A deeper exploration of these qualitative aspects would provide more insight into the method's practical impact.
- The submission’s structure makes it challenging to follow. Key formulas from Algorithm 1 are located in the appendix, and a large portion of the experimental results is also placed there, leaving only one main results table in the main text.


[1] Laskin M, Yarats D, Liu H, et al. URLB: Unsupervised Reinforcement Learning Benchmark[C]//Thirty-fifth Conference on Neural Information Processing Systems Datasets and Benchmarks Track (Round 2).

**Questions:**

The proposed tighter bound on mutual information is presented as beneficial for skill diversity, but the paper does not provide a formal theorem to support this claim. More rigorous theoretical analysis is necessary.

---

> ### Author Response · Authors · 2024-11-21
> **Rebuttal**
>
> Thank you for your feedback and apologies for the delay.
>
> - Missing proof that Skillset Empowerment measures tighter bound on empowerment
>
> We have added the simple proof in section A of the appendix that Skillset Empowerment learns a tighter bound on the empowerment of a state relative to existing approaches (e.g., DIAYN) that train the variational posterior by minimizing the KL divergence between the posterior of the current skillset and the variational posterior.  We had left this out originally because we felt the wording in the original manuscript was sufficient, but have now added it to the paper.  A tighter bound on empowerment results in part because for any ($\phi,\pi$) skillset learned by existing methods, our approach learns a higher variational mutual information for the same $(\phi,\pi)$ skillset because our approach can learn a more accurate variational posterior $q_{\psi}$.  Our approach may then find a different $(\phi_{SE},\pi_{SE})$ skillset with larger variational mutual information.
>
> - Approach assumes a model of the transition dynamics, which means unsupervised RL is not needed.
>
> A couple of comments here.  First, measuring the empowerment of a state has other uses in addition to solving downstream MDPs.  For instance, empowerment-based RL (Klyubin et al., 2005) in which agents maximize an empowerment reward, requires that agents are able to measure the empowerment of states.
>
> Second, it is true that this approach requires a model but we have already shown how this can be overcome (see paper provided in supplementary materials).  This paper maximizes a principled replacement for the mutual information between skills and states that only requires an off-policy dataset of transitions.  This updated objective matched the performance of Skillset Empowerment in all domains.
>
> - Compute requirements are high
>
> This is true but it is also learning very large skillsets with the high compute.  Our approach also provides a principled way to maximize the mutual information between skills and states because it maximizes a tighter bound on the mutual information.  Previous empowerment-based approaches have required less compute but are also less principled as they only maximize a loose lower bound on mutual information between skills and states.
>
> - Missing qualitative examination of skills
>
> Is there something in particular you had in mind?  In the $H(S_n|Z)$ visualization we show the skill-terminating states targeted by each skill.  Given the state coverage shown in the $H(S_n)$ visual, we are learning skills that target nearly all reachable skill-terminating states.
>
> - Missing downstream task performance
>
> Given that we are trying to maximize mutual information between skills and states, we believe this mutual information is the most important metric and is a metric that many other mutual information maximizing algorithms have not provided (e.g., APS, CIC), which makes it unclear how well these other approaches work.  Also, general purpose agents need to be able to achieve a vast number of tasks so showing good performance on one or two tasks is not illustrative of a widely capable agent.

---

> > ### Comment · Reviewer_jSiL · 2024-11-24
> >
> > Thanks for the reply.
> >
> > The clarification on mutual information as the primary evaluation metric is noted.  My concern regarding the lack of downstream task evaluations remains. Demonstrating performance on downstream tasks is necessary to validate the practical utility of the learned skillset, particularly given the computational demands of your method. While maximizing mutual information provides theoretical insights, it does not directly show how these skills perform in real-world or task-specific scenarios.
> >
> > The argument that downstream evaluations may not capture general-purpose capabilities is acknowledged, but downstream tasks serve as a standard way to demonstrate the applicability of the learned skills. This is especially relevant for your method, which claims to achieve large-scale skill learning at a significant computational cost. Without such evaluations, it is difficult to justify the trade-off between the resource requirements and practical benefits.

---

### Official Review · Reviewer_mbwo · 2024-11-02

**Soundness:** 3
**Presentation:** 3
**Contribution:** 3
**Rating:** 8
**Confidence:** 3

**Summary:**

The paper introduces a new method for mutual information skill learning. The idea is to estimate a tighter mutual information lower-bound, by minimizing the kl divergence between the true skill posterior of the current skillset and the variational posterior of the current skillset, to avoid significant non-stationarity in the traditional unsupervised RL setting, the problem is formulated as an actor-critic bandit problem. The experiments show the advantage of the method over the baselines in stochastic setting.

**Strengths:**

- The idea of conditioning the posterior on the policy and skill distribution is interesting and formalizing the problem as bandit is novel
- Results are strong and outperform the baselines by a large margin, which shows the failure of most methods in stochastic domains.
- The number of skill learned by the method is much larger than the baselines, which shows that there is a big room for improvement in the problem of unsupervised skill learning.
- The visualization are insightful and shows that the method is really learning something.

**Weaknesses:**

- The objective for the existing empowerment skill learning methods is not clear to me (equation 6) some method condition only on the current state while some condition on the terminal state, so it is not clear to me how this notation encompasses these methods, can you add specific examples for using different conditioning and how the notation differs between methods?

- The argument of the loose lower bound is not that rigorous, in the paragraph from line 195-199, it does not say why the posterior differs largely from the posterior of the current skill, is it related to how the posterior is trained? can you give a formal proof showing how/why the posteriors of the candidate skill can differ significantly from the current skill?

- The notation is a bit confusing (equation 8 for example, is $\phi$ in the $\log q_{\psi^*}$ the same as the \phi in $\log \phi (z \mid s_0)$? I guess you meant conditioning on the parameters of $\phi$ right?), I recommend to make the notion clearer for example by using a different symbol than $\phi$ in the $\log q_{\psi^*}$ term in equation 8 and clarify what each symbol represents.

- Having |\pi| critics and posteriors seems a very limiting design choice for complex problems in which a more expressive model is usually needed, could you discuss the scalability of your method to more complex problem?

**Questions:**

- Can also add a visualization of the baselines for qualitative comparison? for example visualizing the entropy of the terminal states distribution in the stochastic four rooms environment.

- Is there any theoretical reason why the skill distribution chosen to be uniform, or is it more of a design choice? how the results will change if you used a different skill distribution? could you discuss this more in the paper?

- Why the results of Ant and Mountain Car are not included in the results table (table 1)? could you include the results ant and mountain car in table 1 and how the results compare to other environments ?

---

> ### Author Response · Authors · 2024-11-22
> **Rebuttal**
>
> Thank you for the feedback and apologies for the delay.
>
> - Argument for loose lower bound is not sufficiently rigorous
>
> Section A in the appendix provides the proof that Skillset Empowerment learns a tighter bound on empowerment than existing approaches that train the posterior to match the true posterior of the current skillset.  The tighter bound on empowerment results in part because the maximum variational mutual information learned by existing approaches must be less than or equal to the variational mutual  information learned by our approach for the same pair of (distribution over skills $\phi$, skill-conditioned policy $\pi$) because our approach learns a more accurate posterior distribution.  In addition, our approach can find a $(\phi,\pi)$ skillset with larger mutual information.
>
> - More precisely defining variational objectives using by existing approaches
>
> Sure we can be more precise here but regardless of whether the posterior is mapping skill-terminating states to skills (e.g., VIC (explicit option version)) or states at each time step to skills (e.g., DIAYN), they both will be loose lower bounds as they are not also conditioned on the skillset distributions $(\phi,\pi)$.
>
> - Confusing notation for distributions
>
> Yes we can improve the notation as $\phi$ both represents the distribution over skills $\phi(z|s_0)$ and a scalar quantity representing the side length of the box that represents the distribution over skills.  In the camera ready, we will switch to describing the distribution over skills as $p(z|s_0,\phi)$, which depends on the parameter $\phi$.
>
> - Having $|\pi|$ critics is not sufficiently expressive
>
> I am not sure I follow.  Each parameter-specific critic approximates how the mutual information of a skillset changes with a small change to one of the parameters in the skill-conditioned policy.  This neural network should be sufficiently expressive to approximate this quantity.
>
> - Baseline visualizations
>
> We did add the state coverage for GCRL in stochastic four rooms (Figure 17) in which the GCRL agent just learns to move to the corners.  We can add the VIC agent as well.  Though because the VIC agents did not learn anything meaningful, there should not be too much to see.
>
> - Why uniform distribution over skills?
>
> Our decision to use a uniform distribution over skills in the shape of a $d$-dimensional cube was done for a few practical reasons.  First, it is easy to compute the entropy of a $d$-dimensional cube as it is just the volume of the cube.  Second, this distribution is defined by a single scalar which simplifies the search problem of finding the mutual information-maximizing $\phi$.  Third, if our approach was to be used for solving downstream tasks, it is simple to use this cube as the action space for the downstream policy that outputs skills.  Fourth, as you are maybe hinting at, the uniform distribution does have the largest entropy.

---

> > ### Author Response · Authors · 2024-11-24
> > **Clarification**
> >
> > If your concern about the parameter-specific critics was that the critics should need to know how the mutual information changes with changes to multiple of the skill-conditioned policy parameters, this is not the case.  The gradient of the mutual information critic $Q(s_0,\pi)$, in which $\pi=f_{\lambda}(s_0)$ (ignoring $\phi$), with respect to $\lambda$ only requires the gradients of the mutual information critic with respect to each dimension of $\pi$.  That is, $\nabla_{\lambda} Q=\nabla_{\pi} Q \cdot \nabla_{\lambda} \pi = \sum_{i=1}^{|\pi|} \nabla_{\pi_i} Q \times \nabla_{\lambda} \pi_i$ (i.e., $\nabla_{\pi} Q \cdot \nabla_{\lambda} \pi$ is the dot product between (i) the gradients of the critic with respect to each parameter in $\pi$ and (ii) the gradients of each parameter in $\pi$ with respect to $\lambda$).  This means that the parameter-specific critics that only measure the changes in mutual information from changes to a single parameter are sufficient to know the gradient with respect to $\lambda$ and the critics do not need to approximate changes to multiple parameters.

---

### Official Review · Reviewer_Mzw9 · 2024-11-04

**Soundness:** 2
**Presentation:** 2
**Contribution:** 2
**Rating:** 3
**Confidence:** 4

**Summary:**

This paper addresses the challenge of developing general-purpose agents capable of executing a wide range of skills in stochastic environments. The authors argue that a key objective for learning a diverse set of skills is to maximize the mutual information between skills and states, which quantifies the variety within the skillset. They critique existing unsupervised methods, such as Empowerment-based skill learning and Unsupervised Goal-Conditioned Reinforcement Learning, for only maximizing loose lower bounds on mutual information, which can limit the learning of diverse skillsets. To overcome this, the paper introduces a new empowerment objective termed "Skillset Empowerment," which aims to maximize a tighter bound on the mutual information. This is achieved by replacing the posterior distribution of the proposed skillset with a variational distribution that is conditioned on the skillset and trained to match the posterior. The optimization problem is formulated as a bandit problem, where actions correspond to skillsets and rewards to the mutual information objective. A novel actor-critic architecture is proposed to solve this bandit problem. Empirical results demonstrate the approach's ability to learn large, abstract skillsets in stochastic domains, including those with high-dimensional observations, outperforming current methods.

**Strengths:**

This paper studies an interesting problem, learning a valuable skill library from a novel perspective.

**Weaknesses:**

I am very concerned about the results of the experiment. The author only conducted experiments on very simple tasks, and did not conduct experiments on complex tasks [1]. I am afraid it is difficult to verify the effectiveness of the algorithm. In addition, the baseline chosen by the author is very old and needs to be compared with some recent progress [2, 3, 4].

[1] Urlb: Unsupervised reinforcement learning benchmark

[2] Controllability-aware unsupervised skill discovery

[3] APS: Active Pretraining with Successor Features

[4] Contrastive Intrinsic Control for Unsupervised Skill Discovery

**Questions:**

Can the author visualize some of the skills learned?

---

> ### Author Response · Authors · 2024-11-22
> **Rebuttal**
>
> - Limited to simple domains
>
> The domains are simple because maximizing the mutual information between skills and states conditioned on a particular distribution over skills $\phi$ and skill-conditioned policy $I(Z;S|\phi,\pi)$ requires a lot of compute.  Approximating this mutual information for a single $(\phi,\pi)$ skillset, let alone many $(\phi,\pi)$, requires sampling a large number of skills from the distribution over skills and executing them in the environment with $\pi$.  So it should not be surprising that we applied our approach to simple domains.  Note that we still did apply our approach to the Brax Ant environment where our agents were able to learn $\sim 20$ million skills despite being compute constrained.
>
> Simple domains are also helpful because we can more easily approximate the mutual information between skills and states learned by our approach and the baselines.  This is something that is missing from prior works in which they provide an algorithm that they claim maximizes the mutual information between skills and states but then do not provide results showing the actual learned mutual information between skills and states (e.g., the algorithms APS and CIC).  They only show results on downstream tasks, but then it is unclear whether the mutual information maximization component is working as intended.
>
> - Missing baselines to CSD, APS, CIC
>
> The purpose of our paper was to target the premise from which all these algorithms were built.  Park et al. state that the purpose of LSD and CSD was in part because they believed $I(Z;S)$ could not be maximized in deterministic settings so an alternative objective was needed.  Liu et al. said APS was designed because they claimed that maximizing $I(Z;S)$ does not encourage exploration.  Laskin et al. designed CIC in part because they believed the issue with maximizing a variational lower bound was that the variational bound should be implemented with contrastive learning.  Our paper shows that these reasons are not correct.  We show that the problem is that the typical variational lower bound objective (e.g., DIAYN) is a loose lower bound on the true mutual information $I(Z;S)$ because the variational distribution $q_{\psi}(z|s_n)$ is not conditioned on the skillset distributions ($\phi,\pi$) .  Our empirical work, in which we are able to successfully measure empowerment in a variety of deterministic and stochastic settings, shows that unlike the claims of the above authors (i) empowerment can be measured in deterministic settings, (ii) exploration is built in to the objective (provided a simulator is available, though we address this limitation in the paper in the supplementary materials), and (iii) a variational mutual information can be maximized without contrastive learning.
>
> - Visualize Skills
>
> The entropy visualizations and in particular the $H(S_n|Z)$ visual show the skill-terminating states targeted by specific skills $z$.  The $H(S_n)$ visual shows the distribution of skill-terminating states targeted by 1,000 randomly sampled skills.  The behavior of the agent in the intermediate time steps should not be relevant.

---

> ### Comment · Reviewer_Mzw9 · 2024-11-25
>
> Thanks for your replay. I am still concerned about the experiments, and I keep my score.

---

### Meta-Review · Area_Chair_7M2h · 2024-12-20

**Metareview:**

The paper introduces a new method for mutual information skill learning that estimates a  tighter mutual information lower-bound by minimizing the KL divergence between the true skill posterior and the variational posterior. The experiments show the method outperforms prior approaches in stochastic settings. Reviewers appreciated the importance of the problem setting, the novelty of the approach, and the strong empirical empowerment relative to prior work. Reviewers also noted that the method may be widely applicable. Reviewers had some questions on the definition of the empowerment objective (mbwo, HA6E), on the tightness of the bound (i.e., is the posterior actually different for different skills) (mbwo) Reviewers also had questions about scalability (mbwo), whether the method requires access to the transition dynamics (jSiL). Reviewers also suggested that the paper might be improved by evaluating on more complex tasks (Mzw9), on transfer to new tasks (jSiL), and include more recent methods in the baselines (Mzw9). Reviewers also made recommendations regarding paper presentation, such as  including a qualitative evaluation of the learned skills (jSiL).

In summary, while the paper proposing a compelling approach to an important problem, the numerous areas for improvement compel me to recommend that the paper be rejected and resubmitted to a future conference. It seems like perhaps the two main areas to focus on for improvement are (1) evaluation on more complex domains with more recent baselines, and (2) lifting the assumption of access to a dynamics model.

**Additional Comments On Reviewer Discussion:**

During the rebuttal, the authors clarified the correctness of their approach (see Appendix A), added new visualizations of the skills (Fig 17). The authors also noted that several of the suggestions were at least partially included in the original submission (e.g., visualization of skills, moderately-complex tasks). Nonetheless, the authors acknowledged that their method does require an accurate model of the dynamics and that it does have high computational demands.

---

### Decision · Program_Chairs · 2025-01-22

Reject